# Particle flux-gradient relationships in the high Arctic: Emission and deposition patterns across three surface types

Theresa Mathes<sup>1</sup>, Heather Guy<sup>2, 3</sup>, John Prytherch<sup>4</sup>, Julia Kojoj<sup>6</sup>, Ian Brooks<sup>3</sup>, Sonja Murto<sup>5</sup>, Paul Zieger<sup>6</sup>, Birgit Wehner<sup>7</sup>, Michael Tjernström<sup>5</sup>, and Andreas Held<sup>1</sup>

<sup>1</sup>Chair of Environmental Chemistry and Air Research, Technische Universität Berlin, Berlin, Germany

<sup>2</sup>National Centre for Atmospheric Science, Leeds, U.K.

<sup>3</sup>School of Earth and Environment, University of Leeds, Leeds, U.K.

<sup>4</sup>Department of Earth Sciences, Uppsala University, Uppsala, Sweden

<sup>5</sup>Department of Meteorology, Stockholm University and Bolin Centre for Climate Research, Stockholm, Sweden

<sup>6</sup>Department of Environmental Science, Stockholm University and Bolin Centre for Climate Research, Stockholm, Sweden

<sup>7</sup>Department of Atmospheric Microphysics, Leibniz Institute for Tropospheric Research, Leipzig, Germany

**Correspondence:** Theresa Mathes (mathes@tu-berlin.de)

**Abstract.** The Arctic is experiencing a warming much faster than the global average, and aerosol-cloud-sea-ice interactions are considered to be one of the key features of the Arctic climate system. It is therefore crucial to identify particle sources and sinks to study their impact on cloud formation and cloud properties in the Arctic. Near-surface particle and sensible heat fluxes were measured using the gradient method during the ARTofMELT Arctic Ocean Expedition 2023. A gradient system was deployed to calculate sensible heat and particle fluxes over three different surface conditions: *wide lead*, *narrow lead*, and *closed ice*. To evaluate the gradient measurements, sensible heat fluxes and friction velocities were compared with eddy covariance data. The strongest mean sensible heat fluxes, ranging from 16 W m<sup>-2</sup> to 51 W m<sup>-2</sup>, were observed over wide lead surfaces, aligning with measurements from the icebreaker. In contrast, *closed ice* surfaces had weak, often negative sensible heat fluxes. *Wide leads* acted as a particle source, with median net particle emission fluxes of 0.09 10<sup>6</sup> m<sup>-2</sup> s<sup>-1</sup>. *Narrow lead* surfaces exhibited both net emission and net deposition, though the particle fluxes were weaker. *Closed ice* surfaces acted as a particle sink, with normalized fluxes around 0.06 cm s<sup>-1</sup>. The gradient method was found to be effective for measuring both sensible heat and particle fluxes, allowing flexible deployment over different surface types. This study addresses the critical need for improved quantification of vertical turbulent particle fluxes and related processes that influence the local particle number budget in the high Arctic.

## 15 1 Introduction

The Arctic is experiencing a warming four times faster than the global average due to increased emissions of greenhouse gases, particles, and other climate drivers (Rantanen et al., 2022; Meredith et al., 2019). This phenomenon, known as Arctic amplification (Wendisch et al., 2023), is driven by various feedback and forcing mechanisms, e.g. the ice-albedo feedback (Hall, 2004) and the long-wave greenhouse effect of low-level mixed-phase clouds (Wendisch et al., 2019). Short-lived climate forcers like methane, tropospheric ozone, and aerosols also contribute to this enhanced change (Arctic Monitoring and Assessment Pro-

gramme, 2021). Atmospheric aerosols exert an influence on the Earth's surface energy budget through two distinct effects. The direct effect involves scattering and absorption of incoming solar radiation (Charlson et al., 1991). The indirect effect occurs when particles act as cloud condensation nuclei or ice nuclei particles, leading to the formation of clouds and changes in cloud properties (Twomey, 1977; Albrecht, 1989). In the summertime high Arctic, particle number concentrations can be 25 occasionally too low to facilitate cloud formation (Mauritsen et al., 2011; Stevens et al., 2018). Hence, it can be concluded that even small anthropogenic or natural sources of aerosols can exert a considerable influence on cloud cover, surface temperature, ice melt or freezing and thus on Arctic climate. It is therefore essential to gain a more detailed understanding of natural Arctic aerosol emissions, aerosol evolution and transport, and the effects on cloud microphysics (Schmale et al., 2021). There is still considerable uncertainty regarding the composition and formation pathways of high Arctic aerosol (Schmale et al., 2021; 30 Lawler et al., 2021). Climate projections of future warming may be oversimplified and even miss processes, in part because of uncertainties about aerosols and their role in cloud formation (Chylek et al., 2016). Furthermore, there is a scarcity of turbulent flux measurements of particles and aerosol precursor gases in Arctic regions, which are crucial for constraining models of Arctic aerosol formation and emission (Willis et al., 2018; Schmale et al., 2021). It is therefore of great importance to improve the quantification of vertical turbulent particle fluxes and other processes contributing to the local particle number budget in 35 the central Arctic region (Saylor et al., 2022).

The characterization of aerosol fluxes over sea ice, especially in the higher Arctic remains incomplete, both in terms of 40 observations and models (Schmale et al., 2021). Scott and Levin (1972) were the first to identify leads in the Arctic as a local source of atmospheric particles. The first quantification of aerosol fluxes over sea ice in the higher Arctic was conducted by Nilsson and Rannik (2001) and Nilsson et al. (2001), and later by Held et al. (2011a, b). Notable flux measurements in the 45 higher Arctic on land have been carried out by Donato et al. (2023), while Grönlund et al. (2002) and Contini et al. (2010) have made particle flux measurements in Antarctica. Nevertheless, very few measurements of aerosol fluxes on the sea ice have been carried out and additional flux measurements are needed to quantify particle sources and sinks. Difficulties arise 50 particularly from the need for particle measurements with a high temporal resolution of ideally 10 Hz, which is essential for flux measurements to capture the fine-scale fluctuations of small variations, combined with the challenge of low particle concentrations. In addition to the first order time response of the device, fluctuation dampening of the inlet must also be taken into account, which is often the limiting factor (Conte et al., 2018). An alternative method for measuring fluxes in challenging 55 environmental conditions, such as the sea ice, is the gradient method, which is based on the theory of flux-profile relationships (Held et al., 2011b; Farmer et al., 2021). The gradient method has been applied in different areas. On moving platforms like ships, the gradient method was applied e.g. by Petelski (2003), Petelski et al. (2005) and Petelski and Piskozub (2006). Multi-year measurements using the gradient method have been performed, for example, by Markuszewski et al. (2024). Assuming that turbulent fluxes are constant with height in the surface layer, the fluxes of momentum, heat and particles can be determined from vertical gradients of the mean quantities of wind speed, temperature and particle number concentration close to the surface (Foken and Mauder, 2024).

The present study is focused on near-surface particle and sensible heat fluxes, which were measured using a gradient system on

ice floes in the Arctic spring. Variations in turbulent fluxes are examined in the context of surface characteristics, meteorological conditions, and particle number concentrations. Our results can be used to identify turbulent particle exchange under the influence of different surface types and help to constrain Arctic aerosol sources and sinks in climate models.

## 2 Data and Methods

This study is based on observations collected during the ARTofMELT (Atmospheric rivers and the onset of Arctic sea ice melt) expedition in the Arctic Ocean in spring 2023 on the Swedish icebreaker Oden. The overarching goal of the expedition was to investigate the processes leading to the onset of sea ice melt and to explore its links to narrow filaments of poleward moving warm and moist air, so-called atmospheric rivers (Tjernström and Zieger, 2025; Secretariat, 2024). Oden departed from Svalbard on 8 May and returned back to Svalbard on 15 June 2023. Particle flux and sensible heat flux data collected 65 during two ice camps will be presented here. The first ice camp took place from 16 May to 21 May on an ice floe at 79.6° N, 1.3° W and the second from 29 May to 11 June at 79.8° N, 2.8° E. Additional data analyzed in this study are derived from a wide range of measurements on board Oden, including: 6-hourly radiosoundings, friction velocities and sensible heat fluxes by eddy covariance and aerosol physical and chemical properties. Additional measurements were made on the ice floe, including friction velocity and sensible heat fluxes by eddy covariance and up and down-welling radiative fluxes.


Profile measurements with the gradient setup, hereafter referred to as *gradient system*, were carried out during the ARTofMELT ice camps on 16 days in total (Table 1). The runtime per day varied from 2.5 hours to 18 hours, depending on access to the ice floe to change batteries; this was sometimes limited because of weather conditions or polar bears in the vicinity. The gradient data has been divided into three categories based on the upstream surface type. These categories are:

1) *Wide lead*: extensive open water surface (influence area dominated by open water, Fig. A1 (a)).  
2) *Narrow lead*: open water within leads within pack ice (mixed influence area affected by open water and ice, Fig. A1 (b)).  
3) *Closed ice*: smooth, snow-covered ice surfaces without ridges or melt ponds (Fig. A1 (c)).

From 17 May to 18 May, the *gradient system* was set up approximately 5 to 10 meters away from a *wide lead*. From 20 May to 21 May, the *gradient system* was close (around 20 m) to a *narrow lead*. From 29 May to 6 June, the *gradient system* was 80 surrounded by a smooth, snow-covered *closed ice* surface, and from 7 June to 9 June again located near a *narrow lead* (see Table 1).

### 2.1 Sampling methodology

To measure near-surface particle concentration profiles we used a 1.5 m high linear actuator (Fig. 1 (a)) with a sensor board (Fig. 1 (b)). The sensor board contained I) an ultrasonic distance sensor (HC-SR04, AZ Delivery, Deggendorf, DE) to measure the 85 distance to the surface with a sampling rate of  $0.5 \text{ s}^{-1}$ , (II) a PT100 precision resistance thermometer (Electrotherm, Geraberg, Germany) for air temperature measurements and (III) a simple hot wire anemometer for measuring wind speed and temperature (Rev. P, Modern Device, Brooklyn, NY, USA), both with a sampling rate of  $1 \text{ s}^{-1}$ . In addition to the in-situ sensors, a 20 cm

long, downward-facing stainless steel inlet (IV) with an inner diameter of 3 mm was mounted on the sensor board. The inlet was connected to a condensation particle counter (CPC 3007, TSI, St. Paul, MN, USA) with 205 cm long conductive silicon tubing with an inner diameter of 4.2 mm. The battery-operated CPC measured the total particle number concentration with a lower cut-off diameter of 10 nm, a temporal resolution of  $1\text{ s}^{-1}$  and a sample volume flow rate of 0.7 l/min. Isopropanol was used as working fluid for the CPC. The linear actuator and the sensors were powered by batteries. A laptop for system control and data recording, batteries and the CPC were placed downwind of the *gradient system* in an aluminum box for weather protection. The *gradient system* was oriented into the wind, upwind of the ship. A tripod was used to stabilize the system and was positioned downwind of the *gradient system* as long as the wind did not change direction during the measurement. A white plastic tarp served as the base for the tripod to prevent sinking due to melting snow. Three additional PT100 resistance thermometers were mounted on the tripod at heights of 8 cm, 90 cm and 120 cm above the surface. These air temperature measurements were used for quality control of the moving PT100 sensor. For wind direction measurements, a small ultrasonic anemometer with a response time of 10 Hz (LI-550F TriSonica Mini, LI-COR Lincoln, NE, USA) was mounted at the top of the tripod on a horizontally aligned arm, 38 cm from the tripod. Data from a 90-degree wind direction sector centered on the instrument tripod and box with batteries and logging system is excluded from the analysis because of flow distortion generated by these obstacles. The use of snow scooters or other potentially polluting activities were minimized during measurement periods. Measurement periods affected by pollution were determined by rapidly increasing particle concentrations and were excluded from the analysis.

105

The gradient system was programmed to approach six different height levels consecutively, starting from the bottom with a distance of 5 cm between the inlet and the surface, followed by distances of 10 cm, 17 cm, 32 cm, 67 cm, and 129 cm. These heights varied by  $\pm 1$  cm depending on the measurement and surface conditions. During descent from 129 cm to 5 cm one additional measurement was taken at 32 cm above the surface, serving as an additional reference to track changes in background particle number concentrations and was not included in the gradient calculations. Height levels were changed every 50 s, resulting in a new profile cycle every 350 s.

In addition to the profile measurements with the *gradient system*, continuous eddy covariance sensible heat flux measurements (Prytherch et al., 2024) were carried out with sonic anemometers on a mast on the foredeck of Oden at a height of 20.3 m (referred to as *ship mast*, Sonic: Metek u-sonic 3) and on a mast on the ice (Guy et al., 2024) at a height of 4 m (referred to as *ice mast*, Sonic: Metek USA-100). Eddy covariance calculations followed standard procedures including despiking, double coordinate rotation and linear detrending. Motion correction and a correction to compensate for flow distortion were also applied for the *ship mast* data (Prytherch et al., 2015). The additional eddy covariance measurements enable a comparison between the sensible heat fluxes and friction velocities ( $u^*$ ) of the different setups. The sensible heat fluxes and friction velocities for the sites *ship mast* and *ice mast* were averaged over 20 minutes. For direct comparison, the data of the *gradient system* was also averaged over 20 minutes. The meteorological data for the periods in Table 1 were measured on the 7th deck of Oden at approximately 25 m height above sea level (Murto et al., 2024). General meteorological conditions during the AoM expedition

were categorized in six periods, mainly defined using temperature and humidity profiles from 6-hourly radio soundings as well as 7-day back-trajectories, calculated with the Lagrangian Analysis Tool LAGRANTO on ERA5 data (see also Table 1; Murto 125 et al. 2024; Murto and Tjernström 2024). Radiation was measured at the *ice mast* site, from a radiometer stand at approximately 40 m from the *ice mast* (Guy et al., 2024).

**Figure 1.** Setup (a) of the near-surface *gradient system* and sensor board (b): (I) distance sensor, (II) PT100 precision resistance thermometer, (III) hot wire anemometer and the (IV) aerosol inlet. The sensor board was attached to the linear actuator.

## 2.2 Data Analysis

Firstly, the height level of the *gradient system* was checked using the distance sensor data. The profile was used for further analysis if more than 80 % of each height data measured by the distance sensor were within  $\pm 20$  % of the set heights. In order

to take into account the response time of all sensors as well as the traveling time of the *gradient system* to the next height level, the first 20 s of each 50 s measurement interval at each height level were discarded. The CPC fully adjusted to a new particle number concentration within 9 s, and the maximum traveling time to the next height level was 10 s. Therefore, only the last 30 s of the measured values per height level were used for further data analysis. Secondly, outliers were removed if values 135 were outside 1.5 times the interquartile range of a moving 20-minute window. The 20-minute interval was chosen to avoid over-emphasizing short-term fluctuations, while still taking into account longer-term changes. The arithmetic mean over 30 s for each height level was calculated for further analysis. Third, the wind speed data measured by the hot wire anemometer and the small ultrasonic anemometer as well as the temperature data measured by the PT100 precision resistance thermometer and the hot wire thermocouple were compared. For the wind, we assume a logarithmic profile with height. By linear regression of

wind speed and logarithmic height, the measurements from the hot wire anemometer were extrapolated to 170 cm, the height  
of the sonic anemometer. For temperature, the sensors for comparison were already at the same heights as the profile measurements. Based on the calculated offsets between the sonic anemometer or PT100 precision resistance thermometer and the hot wire thermocouple, the data were adjusted so that only the hot wire anemometer data were used for the following analyses.

In the last step, the measured particle number concentrations were corrected for particle losses inside the tubing and inlet. In  
particular, small particles are lost by diffusion to tubing and inlet walls (Kulkarni et al., 2011). The diffusional losses were calculated following Gormley and Kennedy (1949). To determine the penetrating fraction as a function of particle size, the particle number size distribution between 15 and 792 nm with a temporal resolution of 12 min, measured with a differential mobility particle sizer (DMPS) on the 4th deck of Oden was used. The inlet was a 4 m long heated whole air inlet located approximately 25 m above sea level. Periods influenced by pollution from the ship's exhaust were excluded based on the derivative of the  
total particle number, following a similar principle as described by Beck et al. (2022). The data were corrected for losses due to diffusion, sedimentation, and inertial deposition using the loss calculator of Von Der Weiden et al. (2009). It was assumed that the particle size distribution determined with the DMPS was representative of the size distribution entering the *gradient system*, given the absence of these measurements on the ice. For each size channel and the inlet characteristics described in Sect. 2.1, the loss rate was determined. The size-dependent loss rates per size channel were then multiplied by the normalized  
size distribution data and the total particle penetration efficiency was estimated by the sum of the adjusted, normalized DMPS data. The hourly median penetrating particle fraction was found to be between 90 % and 98 %, with an overall median of 96 %. This variation is to be expected due to the strong dependence on the size distribution (Kulkarni et al., 2011). To correct for diffusional particle losses the measured particle number concentration of the *gradient system* was divided by the penetrating particle fraction value.


Evaluating the gradients from ascending stepped profiles has a potential source of bias if the particle number concentration, wind speed and temperature change strongly during one full profile cycle of 350 s. Therefore, the particle and sensible heat flux were evaluated as mean trends for intervals of 20 minutes. For this purpose, 20-minute intervals were created. For each  
20-minute interval, the median time point was determined and it was checked that at least five of the six heights within the 20 minutes were correctly measured by the *gradient system*. Linear interpolation based on time was used to determine the corresponding value at the median time point for the three measurements of particle number concentration, wind speed and temperature. The data analysis performed is shown step by step in Figure A2.

### 2.3 Flux-profile relationships

Turbulent fluxes are considered to be constant with height in the surface layer (Holton and Hakim, 2013). Following the Monin-  
Obukhov similarity theory (Foken and Mauder, 2024), flux-gradient relationships for momentum flux, sensible heat flux ( $H$ )

and particle flux ( $P$ ) are:

$$\overline{U'w'} = -K_m \frac{\partial \bar{U}}{\partial z} \quad (1)$$

$$H = \overline{w'T'} = -K_H \frac{\partial \bar{T}}{\partial z} \quad (2)$$


$$P = \overline{w'C'} = -K_P \frac{\partial \bar{C}}{\partial z} \quad (3)$$

where the momentum flux  $\overline{U'w'}$  is the covariance of turbulent fluctuations of the horizontal wind speed,  $U$  ( $\text{m s}^{-1}$ ) and vertical velocity,  $w$  ( $\text{m s}^{-1}$ ).  $K_m$  is the turbulent diffusion coefficient for momentum. The sensible heat flux ( $H$ )  $\overline{w'T'}$  is the covariance of the turbulent fluctuations of vertical wind speed  $w$  and temperature (K) and  $K_H$  is the turbulent diffusion coefficient for sensible heat. To convert the kinematic sensible heat flux to units  $\text{W m}^{-2}$  we multiplied the value in units  $\text{K ms}^{-1}$  with the density of air ( $1.225 \text{ kg m}^{-3}$ ) and the specific heat capacity ( $1005 \text{ J kg}^{-1} \text{ K}^{-1}$ ) of air. The particle flux ( $P$ )  $\overline{w'C'}$  is the covariance of the turbulent fluctuations of vertical wind speed  $w$  and particle number concentration  $C$  ( $\text{cm}^{-3}$ ) and  $K_P$  is the turbulent diffusion coefficient for particles. The measurement height is  $z$  (m). Based on the mixing length model (Foken and Mauder, 2024),  $K_m$  is often parameterized using:

$$K_m = \kappa \cdot z \cdot u^* \quad (4)$$

$\kappa$  is the von Karman constant ( $= 0.40$ ),  $z$  is the measurement height, and  $u^*$  is the friction velocity, which can be derived from the wind profile measurements according to Equation 5:

$$u^* = \sqrt{-\overline{U'w'}} = \kappa \cdot z \cdot \frac{\partial U}{\partial \ln z} = \kappa \frac{\partial U}{\partial \ln z} \quad (5)$$

In previous studies, values between 1.0 and 1.39 were found for the ratio  $K_H / K_m$  (e.g. Foken and Mauder (2024)). Assuming 190  $K_m = K_H = K_P$  (e.g. Wieringa 1980; Held et al. 2011b) the sensible heat flux ( $H$ ) and the particle flux ( $P$ ) are obtained from the profile measurements by:

$$H = \overline{w'T'} = -\kappa \cdot u^* \cdot \frac{\partial T}{\partial \ln z} \quad (6)$$

$$P = \overline{w'C'} = -\kappa \cdot u^* \cdot \frac{\partial C}{\partial \ln z} \quad (7)$$

The friction velocity  $u^*$  was determined from the measured gradient of the wind speed (Equation 5). The gradient was calculated from a linear regression of the five wind speeds against the five logarithmic heights of each profile. The temperature  $T$  and particle concentration  $C$  gradients were also determined by linear regression to calculate the sensible heat flux ( $H$ ) and the particle flux ( $P$ ). Figure A3 shows an example of measured particle concentration (a) and wind speed profiles (b) and the linear regression fits. The sensible heat flux ( $H$ ) is defined as positive or upward if  $T$  decreases with increasing height. A 200 positive particle flux ( $P$ ) is defined as upward when  $C$  decreases with increasing height.

The calculation of fluxes from the mean gradients as described above assumes neutral stability; for stable or unstable stratification, stability corrections are required (Foken and Mauder, 2024). The need for a stability correction can be evaluated via the stability parameter  $z/L$ , where  $L$  is the Obukhov length:

$$L = -\frac{u^{*3}}{\kappa \frac{g}{T} w' T'} \quad (8)$$

where  $g$  is the gravitational acceleration ( $= 9.81 \text{ ms}^{-2}$ ), and  $T$  is the temperature. Positive values for the stability parameter  $z/L$  indicate stable stratification, negative values refer to unstable conditions, and neutral stratification is given by a value close to 0. To test whether the assumption of neutral stratification is justified,  $z/L$  values were calculated for the *gradient system* and indicated for the interquartile range values of  $-0.0024 < z/L < 0.0357$ . Therefore, a majority of the measurements were taken under near-neutral conditions and we can ignore stability correction functions for Equations 5 - 7 and accept a  $< 9\%$  error.


In the case of stable stratification, strong temperature gradients can occur very close to the surface. In order to determine periods with possible vertical decoupling, a decoupling metric based on the Brunt-Väisälä frequency has been calculated (Foken, 2023; Peltola et al., 2021). While the decoupling metric indicates coupled or weakly coupled periods when measuring at the *wide lead*, decoupling is frequently possible when measuring over *closed ice* surfaces. In order to exclude periods with weak turbulence and thus a high probability of vertical decoupling, the fluxes calculated from the *gradient system* were classified according to  $u^*$ . For further discussion of the turbulent particle and sensible heat fluxes according to the flux-profile relationships, all mean and median values as well as the number of intervals are based on periods when  $u^* \geq 0.15 \text{ m s}^{-1}$ .


Applying the profile equation for  $u^*$  (Equation 5) from a height  $z_0$ , at which the horizontal wind speed of the extrapolated logarithmic wind profile becomes zero, to the height  $z$ , gives:

$$U(z) - U(z_0) = U(z) = \frac{u^*}{\kappa} \cdot \ln \frac{z}{z_0} \quad (9)$$

The roughness length  $z_0$  is determined by rearranging Equation 9:

$$z_0 = \exp\left(\ln(z) - \frac{\kappa U}{u^*}\right) \quad (10)$$


To allow interpretation of particle fluxes independent of varying particle number concentrations and with other studies, a normalized flux  $V_D$  [ $\text{cm s}^{-1}$ ], is calculated by Equation 11:

$$V_D = -P/C \quad (11)$$

For reasons of comparability with other studies and the often used deposition velocity, a positive normalized flux indicates a net deposition flux. To emphasize that we do not only refer to deposition processes we use the term normalized flux for  $V_D$ .


Notably, the relative measurement uncertainties are large for small fluxes, which is common in the Arctic due to low particle number concentrations. The mean standard deviation (SD) of the  $1 \text{ s}^{-1}$  measurements at a given height was highest for the

particle number concentration, ranging from 3 % to 21 %, followed by temperature (ranging from 1 % to 10 %) and wind speed (ranging from 2 % to 7 %). The standard deviation was calculated for each individual height level; following, the mean standard deviation across all height levels for one day is used. Monte Carlo simulations were utilized to estimate the uncertainty of the flux estimate due to the measurement uncertainty. For each profile, 10,000 values, within the mean standard deviation of that day around the mean value, were randomly generated for each height level, based on the measured value for particle number concentration, temperature and wind speed. The fluxes were then determined 10,000 times with the ensemble of randomly generated values for temperature and particle number concentration, as well as the  $u^*$  calculated from the wind speed gradient. The 90th percentile of the Monte Carlo simulated fluxes was calculated for each profile (Watanabe and Pfeiffer, 2022). The median of the 90th percentiles for one day was then used as the overall uncertainty (maximum error value) for that day. The median errors are 0.9 (SD  $\pm$  2.2)  $\text{W m}^{-2}$  for the sensible heat flux and 0.27 (SD  $\pm$  0.38)  $10^6 \text{ m}^{-2} \text{s}^{-1}$  for the particle flux.

### 3 Results and Discussion

The particle concentrations varied on the different measurement days with different surface types. During the phase when the *gradient system* was influenced by *wide leads* the mean particle number concentration was around  $197 \text{ cm}^{-3}$  (see Table 1). For the *closed ice* measurement phase the concentration was between 10 and  $369 \text{ cm}^{-3}$ . When the measurements were influenced by *narrow leads* the mean particle number concentration was between 77 and  $122 \text{ cm}^{-3}$ . In the meteorological period 4, characterized by cool and dry northerly winds, the particle number concentration was notably low at  $10 \text{ cm}^{-3}$  and  $37 \text{ cm}^{-3}$ . The particle number concentration also decreased in the transition from period 2 to period 3 from 18 to 20 May. This was characterized by a change in wind direction from north / south-westerlies to southerly winds and mild and moist conditions, which resulted in a temporary warming period. Prior to the onset of melting on 10 June, northerly wind brought air masses with low particle number concentration and mainly originated from over the pack ice (Murto and Tjernström, 2024). Wind from the west or east may indicate influence from terrestrial areas like Greenland or Svalbard, respectively.

For the *wide lead*, *closed ice* and *narrow lead* surface types, different values of the surface roughness length ( $z_0$ ) are expected. From Equation 10, the highest value of  $z_0$  was found for the *wide lead* surface type with a median of  $4.9 \cdot 10^{-3} \text{ m}$ . Comparable values of lead areas between sea ice / polynyas show lower  $z_0$  values of e.g.  $0.1$  to  $2.2 \cdot 10^{-3} \text{ m}$  (Held et al., 2011a) or  $0.6$  to  $0.9 \cdot 10^{-3} \text{ m}$  (Nilsson and Rannik, 2001). Differences may be due to different footprints, which are not entirely influenced by leads. The lowest  $z_0$  was found for the *closed ice* surface type with a median value of  $0.3 \cdot 10^{-3} \text{ m}$ . These results are in agreement with the  $z_0$  values determined in the high Arctic by Nilsson and Rannik (2001) and Persson et al. (2002), which indicate a flat ice surface unaffected by other factors. The mixed *narrow lead* surface type had  $z_0$  values around  $1 \cdot 10^{-3} \text{ m}$ , which also agrees with measurements by Held et al. (2011a); Andreas et al. (2010) and Nilsson and Rannik (2001).

**Table 1.** Overview of the measurement phases from 17 May to 10 June 2023 on various surface types. The start and end times describe the measurement period, although this may be interrupted for several hours by battery power failures and may continue into the next day. The meteorological periods (2 = cold, north westerlies / south westerlies; 3 = mild and humid, first temporary warming, southerly winds; 4 = cooler and drier, mostly northerly winds, 5 = almost melting, warm and dry air at altitude, west winds) describe the meteorological conditions, determined for the entire expedition based on radiosonde data, trajectories, and wind direction, wind speed (Murto et al., 2024; Murto and Tjernström, 2024). The column *Conc* describes the median particle number concentration (independent of the  $u^* \geq 0.15 \text{ m s}^{-1}$  filter) with standard deviation and the final four columns show how many 20-minute intervals of that day were dominated by net positive (upward) and net negative (downward) flux measurements for sensible heat flux (*H*) and particle flux (*P*). The values in brackets refer to the total number, the values outside the brackets refer to intervals with  $u^* \geq 0.15 \text{ m s}^{-1}$ .

| Surface type | Date   | Start time | End time | Period | Conc                               | <i>H N<sub>neg</sub></i> | <i>H N<sub>pos</sub></i> | <i>P N<sub>neg</sub></i> | <i>P N<sub>pos</sub></i> |
|--------------|--------|------------|----------|--------|------------------------------------|--------------------------|--------------------------|--------------------------|--------------------------|
| Wide lead    | 17.05. | 13:40      | 16:30    | 2      | 189 ( $\pm 2$ ) $\text{cm}^{-3}$   | 0 (0)                    | 8 (8)                    | 0 (0)                    | 8 (8)                    |
| Wide lead    | 18.05. | 15:40      | 18:10    | 2      | 205 ( $\pm 16$ ) $\text{cm}^{-3}$  | 0 (0)                    | 7 (7)                    | 3 (3)                    | 4 (4)                    |
| Narrow lead  | 20.05. | 14:00      | 18:30    | 3      | 119 ( $\pm 54$ ) $\text{cm}^{-3}$  | 4 (9)                    | 0 (0)                    | 2 (5)                    | 2 (4)                    |
| Narrow lead  | 21.05. | 10:40      | 13:00    | 3      | 104 ( $\pm 2$ ) $\text{cm}^{-3}$   | 6 (7)                    | 0 (0)                    | 5 (6)                    | 1 (1)                    |
| Closed Ice   | 29.05. | 12:00      | 18:40    | 4      | 369 ( $\pm 75$ ) $\text{cm}^{-3}$  | 11 (13)                  | 1 (3)                    | 10 (12)                  | 2 (4)                    |
| Closed Ice   | 31.05. | 20:40      | 22:50    | 4      | 318 ( $\pm 16$ ) $\text{cm}^{-3}$  | 5 (6)                    | 1 (1)                    | 6 (7)                    | 0 (0)                    |
| Closed Ice   | 01.06. | 11:30      | 00:40    | 4      | 239 ( $\pm 42$ ) $\text{cm}^{-3}$  | 5 (31)                   | 7 (2)                    | 4 (22)                   | 3 (16)                   |
| Closed Ice   | 02.06. | 10:40      | 03:00    | 4      | 143 ( $\pm 57$ ) $\text{cm}^{-3}$  | 4 (24)                   | 6 (12)                   | 9 (25)                   | 1 (11)                   |
| Closed Ice   | 03.06. | 14:00      | 21:00    | 4      | 10 ( $\pm 1$ ) $\text{cm}^{-3}$    | 0 (2)                    | 1 (19)                   | 0 (4)                    | 1 (17)                   |
| Closed Ice   | 04.06. | 13:40      | 03:50    | 4      | 37 ( $\pm 23$ ) $\text{cm}^{-3}$   | 27 (29)                  | 10 (12)                  | 21 (24)                  | 16 (17)                  |
| Closed Ice   | 05.06. | 09:40      | 04:20    | 5      | 272 ( $\pm 147$ ) $\text{cm}^{-3}$ | 6 (24)                   | 0 (14)                   | 6 (36)                   | 0 (2)                    |
| Closed Ice   | 06.06. | 10:20      | 00:20    | 5      | 238 ( $\pm 54$ ) $\text{cm}^{-3}$  | 30 (35)                  | 3 (5)                    | 31 (34)                  | 2 (6)                    |
| Narrow lead  | 07.06. | 10:30      | 23:50    | 5      | 77 ( $\pm 8$ ) $\text{cm}^{-3}$    | 23 (37)                  | 2 (2)                    | 15 (26)                  | 10 (13)                  |
| Narrow lead  | 08.06. | 09:40      | 00:10    | 5      | 77 ( $\pm 21$ ) $\text{cm}^{-3}$   | 15 (37)                  | 2 (6)                    | 11 (20)                  | 6 (23)                   |
| Narrow lead  | 09.06. | 15:20      | 05:40    | 5      | 122 ( $\pm 185$ ) $\text{cm}^{-3}$ | 22 (35)                  | 3 (5)                    | 17 (27)                  | 8 (13)                   |

### 3.1 Friction velocity and sensible heat flux

The gradient method is evaluated against the eddy covariance method by comparing the friction velocity ( $u^*$ ) and the sensible heat flux (*H*) determined by the *gradient system* with the same parameters determined on the ship (*ship mast*) and the ice (*ice mast*) by eddy covariance. It must be kept in mind that different footprints result from the different locations and heights of the measurements at 20 m (*ship mast*), 4 m (*ice mast*) and 1 m (*gradient system*) and their surrounding surface types, and thus, a perfect agreement can not be expected.

**Figure 2.** (a) Scatter plot between friction velocities [ $u^*$ ,  $\text{m s}^{-1}$ ] measured by the *ship mast* (y-axis) and the *ice mast* (x-axis). (b) Scatter plot between friction velocities measured by the *gradient system* (y-axis) and the *ice mast* (x-axis). (c) Scatter plot between friction velocities measured by the *gradient system* (y-axis) and the *ship mast* (x-axis). The dark blue lines symbolize the 1:1 line, the turquoise lines the 50% deviations. The beige line shows the linear regression fit. The three different symbol colors in (b) and (c) divide the measurements according to the surface type, which influenced the *gradient system*.

The absolute differences of the  $u^*$  values are almost always within  $0.1 \text{ m s}^{-1}$  for the whole dataset, and notably, the variation of the difference of  $u^*$  measured on the *ship mast* and on the *ice mast* by eddy covariance is similar to the variation of the differences between the *gradient system* and the eddy covariance systems. The median deviation of  $u^*$  between the *gradient system* and the *ship mast* is  $0.02 \text{ m s}^{-1}$  and between the *gradient system* and the *ice mast* it is  $0.04 \text{ m s}^{-1}$ . The median of the differences between two eddy covariance data sets from *ship mast* and *ice mast* is only slightly smaller at  $0.01 \text{ m s}^{-1}$ . The results demonstrate that the eddy covariance measurements on the *ship mast* and *ice mast* show the highest degree of agreement (Fig. 2 (a)). Nevertheless, as can be seen in Fig. 2 (c), many of the  $u^*$  values determined at the *ice mast* and *gradient system* sites are in good accordance and close to the 1:1 line. In particular, when both systems are influenced by the same surface type (*closed ice*), there is good agreement of  $u^*$ . The agreement of the  $u^*$  values determined at the *ship mast* and *gradient system* sites (Fig. 2 (b)) is slightly lower, but still consistent with expectations. The deviation of  $u^*$  between the *gradient system* and *ship mast* measurements was less than 50% in 63% of the cases. The deviation of  $u^*$  between the *gradient system* and *ice mast* measurements was less than 50% in 77% of the cases, and between the *ship mast* and *ice mast* in 89% of the cases. The comparison includes all data, also when  $u^* 

Figure 3. (a) Scatter plot between sensible heat fluxes [ $H$ ,  $\text{W m}^{-2}$ ] measured by the *ship mast* (y-axis) and the *ice mast* (x-axis). (b) Scatter plot between sensible heat fluxes measured by the *gradient system* (y-axis) and *ship mast* (x-axis). (c) Scatter plot between sensible heat fluxes measured by the *gradient system* (y-axis) and *ice mast* (x-axis). The dark blue lines symbolize the 1:1 line. The three different colors in (b) and (c) divide the measurements according to the surface type, which influenced the *gradient system*.

Figures 4 (a) and (b) show sensible heat fluxes measured with the *gradient system* with a time resolution of 20-minutes, as well as the 20-minute fluxes measured with two eddy covariance systems *ice mast* and *ship mast*. All measured values here are for conditions of  $u^* \geq 0.15 \text{ m s}^{-1}$ . The most pronounced positive sensible heat fluxes were observed on 17 and 18 May by the *gradient system*, with 20-minute mean values of  $61 (\text{SD} \pm 34) \text{ W m}^{-2}$  on 17 May (Fig. 4 (a)) and  $22 (\text{SD} \pm 9) \text{ W m}^{-2}$  on 18

May influenced by the *wide lead* surface type (Fig. 4 (b)). Sensible heat fluxes from the *ship mast*, which was also surrounded by *wide leads*, were also strongly positive on 18 May, with 20-minute mean values of  $49 (\text{SD} \pm 8) \text{ W m}^{-2}$ . A decreasing temperature from the surface (open water  $-1^\circ\text{C}$ ) to the top of the *gradient system* ( $-3$  to  $-2^\circ\text{C}$ ) can explain the positive sensible heat fluxes measured on these days. Similar results, although smaller in magnitude (sensible heat flux of  $9 \text{ W m}^{-2}$ ), have also been observed by Held et al. (2011b).

**Figure 4.** Sensible heat flux [ $H$ ,  $\text{W m}^{-2}$ ] on (a) 17 May and (b) 18 May 2023 from the *gradient system* setup influenced by *wide leads* with a time resolution of 20-minutes as well as sensible heat flux from the *ship mast* and *ice mast* with a time resolution of 20-minutes. (c) Sensible heat flux from the second ice camp from 29 June to 10 June 2023 from the three different measurement setups (*gradient system*, *ice mast*, and *ship mast*). In (a) and (b), all  $u^* \geq 0.15 \text{ m s}^{-1}$ ; in (c), *gradient system* data are shown in different colors depending on  $u^*$ .

For a longer comparison of sensible heat fluxes measured with the eddy covariance and the gradient methods, Fig. 4 (c) shows the sensible heat flux for the different measurement setups for the second ice camp. From 29 May to 5 June, the *gradient system* and the *ice mast* are surrounded by the same surface type (*closed ice*). At the beginning of June, sensible heat fluxes measured by the *gradient system* over the *closed ice* surface were very low. For this period, the low  $u^*$  (beige circles) indicates possible vertical decoupling. On 2 June, the mean net flux estimated from the *gradient system* and the *ice mast* was  $H = -5$  (SD  $\pm 5$ )  $\text{W m}^{-2}$ , whereas  $H = 4$  (SD  $\pm 4$ )  $\text{W m}^{-2}$  were estimated from the *ship mast* respectively. The colder ice surface, in contrast to the open water, leads to a change in the temperature gradient, resulting in lower fluxes. Most comparable studies found low positive values (Held et al., 2011b; Persson et al., 2002), and Sedlar et al. (2011) report mean values between -2 and 6  $\text{W m}^{-2}$ . The difference between the *gradient system* and the *ice mast* / *ship mast* site on 5 June could be attributed to low turbulence, which is indicated by a small  $u^*$  of 0.04 to 0.21  $\text{m s}^{-1}$ . In the final measurement phase from 7 June until 9 June, in 310 which the footprint of the *gradient system* was defined by a mixed surface condition (*narrow lead*) (Table 1), the fluxes of the *gradient system* deviate strongly from those of the other measurement setups. The calculated fluxes for the *gradient system* are negative, with an average flux of -7 (SD  $\pm 5$ )  $\text{W m}^{-2}$  for 7 June and -3 (SD  $\pm 3$ )  $\text{W m}^{-2}$  for 8 June. In contrast, the fluxes at the *ice mast* and *ship mast* locations are positive, at around  $H = 5$  (SD  $\pm 5$ )  $\text{W m}^{-2}$  and  $H = 10$  (SD  $\pm 4$ )  $\text{W m}^{-2}$ . The discrepancy may be attributed to differences in surface conditions between the measurement setups, as well as to variations in 315 the areas of influence resulting from the different measurement heights. It may be hypothesized that the *ship's mast* footprint was characterized by a larger fraction of open water.

Spatially highly variable factors such as cloud cover and the surface type can have a direct and strong influence on the local sensible heat exchange (Sedlar et al., 2011; Held et al., 2011b; Lüers and Bareiss, 2010; Li et al., 2020). Sedlar et al. (2011) demonstrated that the surface energy budget is strongly influenced by the properties of the clouds and the degree of cloud cover. This may also be a potential explanation for the notable differences observed in the results from the three measurement 325 setups, which can be attributed to different upstream conditions for the different measurement sites. Measurements of the total up- and downwelling radiation, as well as the air temperature and the surface brightness temperature at the *ice mast* site (Guy et al., 2024), indicate a highly complex relationship between the surface heat flux and the radiation budget. The downwelling radiative flux of  $950 \text{ W m}^{-2}$  and an upwelling radiative flux of  $900 \text{ W m}^{-2}$  during midday (e.g. 21 May 2023) result in an increase in the surface brightness temperature. This coincides with a positive sensible heat flux around 12:00 UTC, which either decreases slightly or returns to zero overnight. This effect is either reduced or absent when cloud cover is increased. The 330 strong influence of local-scale processes on the sensible heat flux in the Arctic is also emphasized by Liu et al. (2024) and Lüers and Bareiss (2010, 2011), among others.


### 3.2 Particle fluxes

In the following analysis of particle fluxes ( $P$ ), again, a distinction is made between the surfaces *wide lead*, *closed ice* and *narrow lead*. Particle fluxes are not only influenced by the surface type but also by particle number concentrations present in the air (Nilsson and Rannik, 2001) and by the micrometeorological conditions including the development of turbulence and atmospheric stability. Particle number concentration variations observed at a single location are attributable, among other factors, to the influence of varying meteorological conditions. To remove the influence of varying particle number concentrations, 340 the normalized flux (in  $\text{cm s}^{-1}$ , e.g. Farmer et al. 2021) calculated according to Equation 11 will also be presented. It should be emphasized that the observed particle fluxes are a combination of emission and deposition fluxes, i.e. net positive fluxes are dominated by particle emission, and net negative fluxes are dominated by particle deposition.


As outlined in Sect. 2.3, the uncertainties of the calculated fluxes can be considerable. For a general understanding, it is thus instructive to first consider typical properties of the particle flux values as a function of surface type, rather than specific flux values. Table 1 gives an overview of how many 20-minute intervals of each day were net positive (upward) and net negative (downward) flux measurements.

**3.2.1 Wide and narrow leads**

Under the influence of *wide leads* on 17 May (Fig. 5 (a)), all 20-minute intervals are characterized by a positive net particle flux, indicating the net emission of particles (Table 1). On 18 May (Fig. 5 (b)), this is the case for approximately half of the measurement time (Table 1). However, taking into account the estimated maximum uncertainty (grey shading in Fig. 5), fluxes may be positive or negative but with a tendency toward net particle emission. The median value of positive net fluxes is 0.12

$10^6 \text{ m}^{-2} \text{ s}^{-1}$  on 17 May and  $0.02 \cdot 10^6 \text{ m}^{-2} \text{ s}^{-1}$  on 18 May. The normalized flux for the *wide lead* surface type on 17 May ranges from  $-0.09$  to  $-0.04 \text{ cm s}^{-1}$  with a median value in terms of net emission of  $-0.06 \text{ cm s}^{-1}$ .

**Table 2.** Overview of the normalized fluxes of this study and comparable studies depending on the surface type in  $\text{cm s}^{-1}$ . Duan et al. (1988) report the mean value. All other studies report median values. The normalized fluxes of Held et al. (2011a) refer to deposition dominated flux periods.

| Surface type | This study | Contini et al. (2010) | Donato et al. (2023) | Duan et al. (1988) | Grönlund et al. (2002) | Held et al. (2011a) | Held et al. (2011b) | Nilsson and Rannik (2001) |
|--------------|------------|-----------------------|----------------------|--------------------|------------------------|---------------------|---------------------|---------------------------|
| Wide lead    | -0.04      |                       |                      |                    |                        | 0.03 to 0.04        |                     | 0.18                      |
| Narrow lead  | 0.02       |                       |                      |                    |                        | 0.05                | -0.05               | 0.04                      |
| Closed ice   | 0.06       | 0.05                  | 0.03                 | 0.03               | 0.33                   | 0.04 to 0.07        | 0.07                | 0.03                      |

Concerning the surface type *narrow lead*, the first phase from 20 May to 21 May is characterized by both positive and negative fluxes, but mostly net particle deposition with a median net deposition flux of  $-0.16 \cdot 10^6 \text{ m}^{-2} \text{ s}^{-1}$  on 20 May (Fig. 5 (c)) to  $-0.03 \cdot 10^6 \text{ m}^{-2} \text{ s}^{-1}$  on 21 May (Fig. 5 (d)). It is evident that the uncertainty is much smaller than for the *wide lead* site, 355 and there are time periods that can be attributed to a specific flux direction outside of the maximum uncertainty. From 7 June to 9 June (see Fig. A4 (a), (b)), all days are dominated by net particle deposition with a median net particle deposition flux of  $-0.03 \cdot 10^6 \text{ m}^{-2} \text{ s}^{-1}$ ). In total, 50 net deposition and 27 net emission intervals were observed under *narrow lead* conditions (Table 1). However, with both negative and positive particle flux values, the average flux can be close to zero and it is not possible to determine a preferred flux direction on these three days. Rather, both net particle emission and deposition intervals 360 may be observed under *narrow lead* conditions. The median normalized flux for particle deposition on 20 May and 21 May is  $0.80 \text{ cm s}^{-1}$  and  $0.03 \text{ cm s}^{-1}$ , respectively. With regard to particle emission, the median normalized flux is observed to be in the range of  $-0.05 \text{ cm s}^{-1}$  and  $-0.03 \text{ cm s}^{-1}$  on 7 and 8 June, and  $-0.06 \text{ cm s}^{-1}$  on 9 June. The normalized flux at *narrow leads* determined by Held et al. (2011b) is in a similar range from  $-0.008$  to  $-0.1 \text{ cm s}^{-1}$ , but is more dominated by emission processes. The normalized flux determined by Nilsson and Rannik (2001) ranges from  $0.029$  to  $0.065 \text{ cm s}^{-1}$  (Table 2).

**Figure 5.** Particle flux [ $P$ ,  $10^6 \text{ m}^{-2} \text{ s}^{-1}$ ] at surface type *wide lead* on (a) 17 May and (b) 18 May 2023 as well as particle flux at surface type *narrow lead* on (c) 20 May and (d) 21 May 2023. The brown color indicates net particle emission intervals, and the blue color indicates net particle deposition intervals. The beige color indicates intervals, where  $u^* < 0.15 \text{ m s}^{-1}$ . The grey-shaded bars illustrate the maximum error value for the day of measurement, as estimated by Monte Carlo simulations.

Compared to measurements over soil and water, there are few direct measurements of particle dry deposition over snow and ice surfaces (Willis et al., 2018; Emerson et al., 2020). In the literature, little distinction is made between the size of the leads and the fraction of open water. For open leads in the Arctic, Held et al. (2011a) and Nilsson et al. (2001) published comparable measurements. Our results are in agreement with previous studies that particles are emitted from leads (Lapere et al., 2024; Nilsson et al., 2001; Held et al., 2011a). Sometimes, however, net particle deposition at leads was also observed. Held et al. 370 (2011a) found both, net particle deposition and emission from leads with a flux of  $-0.03 \text{ } 10^6 \text{ m}^{-2} \text{ s}^{-1}$  to  $0.02 \text{ } 10^6 \text{ m}^{-2} \text{ s}^{-1}$ . Nilsson and Rannik (2001) also recorded positive and negative fluxes across leads. Positive fluxes in the vicinity of leads had a median value of  $0.032 \text{ } 10^6 \text{ m}^{-2} \text{ s}^{-1}$  and negative fluxes had a median of  $-0.024 \text{ } 10^6 \text{ m}^{-2} \text{ s}^{-1}$  (Nilsson and Rannik, 2001). When both positive and negative fluxes are included, the fluxes are not notably different from zero (Nilsson and Rannik, 2001), which was also observed in this measurement campaign. Held et al. (2011b) calculated fluxes of  $0.006 \text{ } 10^6 \text{ m}^{-2} \text{ s}^{-1}$  to  $0.057 \text{ } 10^6 \text{ m}^{-2} \text{ s}^{-1}$  using the gradient method and fluxes of  $-0.02 \text{ } 10^6 \text{ m}^{-2} \text{ s}^{-1}$  to  $0.05 \text{ } 10^6 \text{ m}^{-2} \text{ s}^{-1}$  using the eddy covariance method in the area affected by leads. Held et al. (2011a) detected an almost equal number of deposition and emission events at leads. 375

Nevertheless, it is clear that the strongest net emission fluxes occur most frequently in the vicinity of sea water (Willis et al., 2018). It has been suggested in the literature that particle emissions detected over sea water consist of sea salt aerosols caused

by wind-driven white caps and bubble bursts at the sea surface (Nilsson et al., 2001). Other studies suggest that the emission  
of particles from narrow leads / wide leads is not exclusively driven by wind (Held et al., 2011a), which is consistent with our  
observations. We could not find a clear correlation between increasing wind speed and a higher net particle emission flux (Fig.  
A6). This is also consistent with the observations of Scott and Levin (1972), who found open lead aerosol production without  
visible bubble activity. They speculated that very small bubbles could still be bursting at the interface between the atmosphere  
and the water and that droplets could be released from the bursting of micro-bubbles during melting or freezing processes.  
Norris et al. (2011) observed bubbles in the surface water of open leads, even during periods of low wind speed and when the  
lead was covered with ice.

### 3.2.2 Closed ice

For eight days, the *gradient system* was under the influence of *closed ice*. Net particle deposition dominated over emission for  
the majority of these eight days. Median net deposition fluxes are around  $-0.02 \text{ } 10^6 \text{ m}^{-2} \text{ s}^{-1}$  with a typical median normalized  
flux of  $0.06 \text{ cm s}^{-1}$ . On 29 (not shown) and 31 May (see Fig. 6 (a)), as well as on 5 and 6 June (Fig. A5 (b)), the particle  
fluxes are negative during nearly the entire measuring interval. However, it is possible that net emission fluxes may occur if  
the maximum estimated uncertainty (grey shading) is taken into account. Net particle emission dominated during one day (3  
June), of which very small fluxes close to zero were observed on 3 June (see A5 (a)) and 4 June. The lowest particle number  
concentrations during the entire expedition were observed on these two days. The median normalized flux was between  $-0.09$   
and  $0.23 \text{ cm s}^{-1}$  on 3 June and between  $-0.07$  and  $0.09 \text{ cm s}^{-1}$  on 4 June. In total, 87 net deposition and 25 net emission  
intervals were observed under *closed ice* conditions (Table 1). Given that the majority of net particle deposition occurs over  
the *closed ice* surface type, it can be concluded that the ice acts mainly as a sink in terms of particle number. Exceptions may  
be attributed to very low particle number concentrations on 3 and 4 June.

400

Net particle emission from a *closed ice* surface like on 1 June (Fig. 6 (b)) has also been observed in other studies, e.g. in  
Nilsson et al. (2001). As particle fluxes and normalized fluxes are strongly dependent on particle size (Nilsson and Rannik,  
2001), a changing size distribution (Fig. 6 (e), (f)) could explain a change from net particle deposition to net particle emission  
405 intervals. Due to Brownian diffusion, the normalized flux of submicron of particles with a diameter of less than 100 nm is  
greater than for particles above 100 nm (Farmer et al., 2021; Nilsson and Rannik, 2001; Grönlund et al., 2002). The particle  
size distribution (measured on the ship) on 31 May shows a clear maximum of particles between 10 and 30 nm (Fig. 6 (e)).  
However, the size distribution is different on 1 June (Fig. 6 (f)); and also on 4 June, not shown). The number of particles in the  
410 10 - 30 nm diameter range decreases, and an increase of larger particles with a diameter of 100 nm or above can be observed. A  
possible explanation for the change in particle size distribution could be a change in wind direction from northeast to northwest  
(Murto et al., 2024). On 1 June, it can be observed that from 17:00 UTC, the concentration of particles in the diameter range  
up to 120 nm increases, and from 19:00 UTC the concentration of particles in the diameter range from 10 - 30 nm decreases  
(Fig. 6 (f)). The net deposition flux before 16:00 UTC changes to a net emission flux after 17:00 UTC (Fig. 6 (b)). Small  
particles with a high normalized flux resulting in a net deposition flux close to the surface become less dominant after 17:00

UTC, which could make previously masked emission processes visible. This effect is not associated with a change in friction velocity (Fig. 6 (d). Possible, albeit weak, particle sources from the *closed ice* surface could be secondary aerosol formed from dimethyl sulfide (Park et al., 2019; Kerminen and Leck, 2001) or organic compounds (Moschos et al., 2022), or fine-mode sea salt aerosol generated by blowing snow (Gong et al., 2023; Yang et al., 2008, 2019). Sea salt aerosols, aerosols from dimethyl sulfide and organic aerosols or are in the size range of 10 to 300 nm (Gong et al., 2023), 50 to 100 nm (Ghahreman et al., 2019) and > 100 nm (Tremblay et al., 2019), respectively, and could thus contribute to the observed particle emission flux. Sea salt aerosols, sublimation from blowing snow as well as secondary organic aerosols are considered more likely as dimethyl sulfide has only been detected in sea water (Ghahreman et al., 2019) or in / near melt ponds (Gourdal et al., 2018), neither of which were present at this site. Wind speeds of more than  $7 \text{ m s}^{-1}$  are required to generate sea salt aerosol from snow covered ice surfaces, (Gong et al., 2023), which were temporarily exceeded in the morning and early afternoon.

Comparing with particle fluxes above closed ice cover in previous studies, a normalized flux between -0.013 and 0.13  $\text{cm s}^{-1}$  was estimated from the gradient method and by eddy covariance during the ASCOS 2008 campaign on an ice floe (Held et al., 2011b) (Table 2). Results from eddy covariance measurements at the Nansen Ice Sheet, Antarctica by Contini et al. (2010), in the MIZ by Held et al. (2011a) and Nilsson and Rannik (2001), over a closed snow surface in Pennsylvania by Duan et al. (1988) and in Ny Ålesund by Donateo et al. (2023) are also within this range. Grönlund et al. (2002) found even higher normalized fluxes over a snow surface in Antarctica with a mean particle normalized flux of  $0.33 \text{ cm s}^{-1}$  (Table 2). Apart from the wide range of normalized fluxes observed, it is clear that the *closed ice* surface acts mainly as a sink for particles. Thus, very few emission-dominated intervals were observed under the influence of *closed ice* and the particle fluxes in these areas are mostly negative, indicating net particle deposition.

**Figure 6.** Overview of surface type *closed ice* on 31 May and 01 June. (a) - (b) Particle flux  $[P, 10^6 \text{ m}^{-2} \text{ s}^{-1}]$ , (c) - (d) normalized flux  $[V_D, \text{ cm s}^{-1}]$  and friction velocity  $[u^*, \text{ m s}^{-1}]$ : The brown color indicates net particle emission, and the blue color indicates net particle deposition. The beige color indicates intervals, where  $u^* < 0.15 \text{ m s}^{-1}$ . The grey-shaded bars illustrate the maximum error value for the day of measurement, as estimated by Monte Carlo simulations. The beige color shows the friction velocity. (e) - (f) Normalized  $dN/d\log Dp$  particle size distribution from 15 to 790 nm (left y-axis) with total particle number concentration, measured with a DMPS on the 4th deck of Oden (white line; right y-axis).

#### 4 Summary and conclusions

Near-surface particle and sensible heat fluxes using the gradient method were successfully determined in the Arctic Ocean as part of the ARTofMELT expedition. A novel battery-powered gradient flux system facilitated measurements at multiple locations by relocating the instrument to quantify turbulent fluxes influenced by three different surface types, i.e. *wide lead*, *narrow lead* and *closed ice*.

The fluxes calculated by the gradient method were validated by comparing sensible heat fluxes and friction velocities measured with two eddy covariance systems. Positive sensible heat fluxes were high at the *wide lead* site, with median fluxes between **16**  $\text{W m}^{-2}$  and **51**  $\text{W m}^{-2}$ , which aligns closely with the eddy covariance flux observed on the *ship mast* ( $49 \text{ W m}^{-2}$ ). For measurements influenced by *closed ice*, sensible heat fluxes were very weak with slightly negative values.

Under the influence of *wide leads*, net particle emission fluxes in the range of  $0.09 \text{ } 10^6 \text{ m}^{-2} \text{ s}^{-1}$  dominated. The observations in this study clearly suggest that open water typically acts as a net particle source. In the presence of *narrow leads*, both net emission and net deposition fluxes were observed. With a median net particle emission flux of  $0.02 \text{ } 10^6 \text{ m}^{-2} \text{ s}^{-1}$  and a median net particle deposition flux of **-0.03**  $10^6 \text{ m}^{-2} \text{ s}^{-1}$ , the particle fluxes are close to zero and less pronounced than under the influence of *wide leads*. The *closed ice* surface, on the other hand, clearly acted as a net particle sink. Median net particle deposition fluxes were around  $-0.1 \text{ } 10^6 \text{ m}^{-2} \text{ s}^{-1}$  with a normalized flux of  $0.06 \text{ cm s}^{-1}$ . In periods with particularly low concentrations, 450 with daily mean particle number concentrations of  $10 \text{ cm}^{-3}$ , normalized fluxes up to **0.09**  $\text{cm s}^{-1}$  were determined, although such low concentrations are associated with high uncertainties. **On several occasions, net particle emission intervals occurred under the influence of closed ice.** These emission events coincided with a shift from small particles in the diameter range of 10 - 30 nm to 100 - 200 nm. Possible emission sources from the ice include, for example, fine-mode sea salt aerosols generated by blowing snow.


Overall, this study provides additional experimental data to corroborate previous findings that different surface types have characteristic effects on the turbulent particle exchange between the atmosphere and the central Arctic Ocean. Furthermore, the 460 gradient method was validated by comparing with simultaneous measurements from three different systems and two different techniques. Quantifying typical particle deposition fluxes to *closed ice* surfaces and typical particle emission fluxes from open leads can reduce the uncertainty in Arctic aerosol models. In order to obtain a representative data set for the models, further measurements are required during different seasons, years and locations. The *gradient system* is an appropriate method for this purpose.

*Data availability.* The data set can be accessed under <https://doi.org/10.1594/PANGAEA.974992>

## Appendix A

A1

**Figure A1.** Three exemplary pictures of the studied surface types: (a) *Wide lead*, (b) *Narrow lead*, (c) *Closed ice*.

**Figure A2.** Step-by-step data analysis procedure for the *gradient system* including temperature (T), wind speed (WS) and particle number concentration (PNC) data

**Figure A3.** Example of vertical profiles of particle concentration [cm<sup>-3</sup>] (a) and wind speed [m s<sup>-1</sup>] (b) over *closed ice* on 5 June 2023. The beige line shows the linear regression fit with  $R^2 = 0.98$  for the particle concentration gradient and  $R^2 = 0.99$  for the wind speed gradient. Utilising the calculated  $u^* \geq 0.15 \text{ m s}^{-1}$ , a flux of  $0.47 \text{ } 10^6 \text{ m}^{-2} \text{ s}^{-1}$  was derived.

**Figure A4.** Particle flux  $[P, 10^6 \text{ m}^{-2} \text{ s}^{-1}]$  at surface type *narrow lead* on (a) 7 June and (b) 8 June 2023. The brown color indicates net particle emission intervals, and the blue color indicates net particle deposition intervals. The beige color indicates intervals, where  $u^* 

**Figure A5.** Particle flux [ $P, 10^6 \text{ m}^{-2} \text{ s}^{-1}$ ] at surface type *closed ice* on (a) 3 June and (b) 6 June 2023. The brown color indicates net particle emission intervals, and the blue color indicates net particle deposition intervals. The beige color indicates intervals, where  $u^* 

**Figure A6.** Particle flux [ $P, 10^6 \text{ m}^{-2} \text{ s}^{-1}$ ] and windspeed [ $\text{m s}^{-1}$ ] at the three surface types *closed ice*, *wide lead* and *narrow lead* for the entire measurement period.

*Author contributions.* TM: conceptualization, methodology, measurements with the gradient system and writing – review and editing with conceptual contributions from HG, JP, JK, IB, SM, PZ, BW, MT and AH. HG and IB provided sensible heat flux and radiation data from the ice mast site. Sensible heat flux data from the ship mast site were provided by JP, SM and MT as well as meteorological data. Particle size distribution data was measured and provided by JK and PZ. Supervision: AH.

*Competing interests.* At least one of the (co-)authors is a member of the editorial board of Atmospheric Chemistry and Physics.

*Acknowledgements.* This work is part of the ARTofMELT (Atmospheric rivers and the onset of Arctic melt) project. The ARTofMELT expedition was supported and organized by the Swedish Polar Research Secretariat (SPRS) on the Swedish research icebreaker Oden in spring 2023 under the SWEDARCTIC program. Support also came from the Swedish Council for Research Infrastructures (Grant 2021-00153) and

the Knut and Alice Wallenberg Foundation (Grant 2016-0024) as well as from the Natural Environment Research Council UK (Grant 475 NE/X000087/1). This study was funded by the Deutsche Forschungsgemeinschaft (DFG, German Research Foundation) – HE5214/10-1, HE5214/11-1 and WE 2757/6-1. The authors are grateful to the co-Chief Scientists Michael Tjernström and Paul Zieger, the SPRS coordinator Åsa Lindgren and the SPRS support team, and to Captain Mattias Petersson and the crew on Oden. The authors would like to thank Max Zeidler, Andre Backhoff and Sven Klemer for technical support. We thank Thomas Foken and an anonymous reviewer for their valuable reviewer comments. I acknowledge the use of *DeepL Write* and *Grammarly* for the preparation of the manuscript.

Albrecht, B. A.: Aerosols, cloud microphysics, and fractional cloudiness, *Science* (New York, N.Y.), 245, 1227–1230, <https://doi.org/10.1126/science.245.4923.1227>, 1989.

Andreas, E. L., Persson, P. O. G., Grachev, A. A., Jordan, R. E., Horst, T. W., Guest, P. S., and Fairall, C. W.: Parameterizing turbulent exchange over sea ice in winter, *Journal of Hydrometeorology*, 11, 87–104, [https://doi.org/https://doi.org/10.1175/2009JHM1102.1](https://doi.org/10.1175/2009JHM1102.1), 2010.

Arctic Monitoring and Assessment Programme, ed.: *AMAP Assessment 2021: Impacts of Short-lived Climate Forcers on Arctic Climate, Air Quality, and Human Health.*, AMAP, Tromsø, Norway, 2021.

Beck, I., Angot, H., Baccarini, A., Dada, L., Quéléver, L., Jokinen, T., Laurila, T., Lampimäki, M., Bukowiecki, N., Boyer, M., Gong, X., Gysel-Beer, M., Petäjä, T., Wang, J., and Schmale, J.: Automated identification of local contamination in remote atmospheric composition time series, *Atmospheric Measurement Techniques*, 15, 4195–4224, <https://doi.org/10.5194/amt-15-4195-2022>, 2022.

Charlson, R. J., Langner, J., Rodhe, H., Leovy, C. B., and Warren, S. G.: Perturbation of the northern hemisphere radiative balance by backscattering from anthropogenic sulfate aerosols\*, *Tellus A: Dynamic Meteorology and Oceanography*, 43, 152, <https://doi.org/10.3402/tellusa.v43i4.11944>, 1991.

Chylek, P., Vogelsang, T. J., Klett, J. D., Hengartner, N., Higdon, D., Lesins, G., and Dubey, M. K.: Indirect Aerosol Effect Increases CMIP5 Models' Projected Arctic Warming, *Journal of Climate*, 29, 1417–1428, <https://doi.org/10.1175/JCLI-D-15-0362.1>, 2016.

Conte, M., Donateo, A., and Contini, D.: Characterisation of particle size distributions and corresponding size-segregated turbulent fluxes simultaneously with CO<sub>2</sub> exchange in an urban area, *Science of The Total Environment*, 622-623, 1067–1078, <https://doi.org/https://doi.org/10.1016/j.scitotenv.2017.12.040>, 2018.

Contini, D., Donateo, A., Belosi, F., Grasso, F. M., Santachiara, G., and Prodi, F.: Deposition velocity of ultrafine particles measured with the Eddy–Correlation Method over the Nansen Ice Sheet (Antarctica), *Journal of Geophysical Research: Oceans*, 115, <https://doi.org/10.1029/2009JD013600>, 2010.

Donateo, A., Pappaccogli, G., Famulari, D., Mazzola, M., Scoto, F., and Decesari, S.: Characterization of size-segregated particles' turbulent flux and deposition velocity by eddy correlation method at an Arctic site, *Atmos. Chem. Phys.*, 23, 7425–7445, <https://doi.org/10.5194/acp-23-7425-2023>, 2023.

Duan, B., Fairall, C. W., and Thomson, D. W.: Eddy Correlation Measurements of the Dry Deposition of Particles in Wintertime, *Journal of Applied Meteorology*, 27, 642–652, 1988.

Emerson, E. W., Hodshire, A. L., DeBolt, H. M., Bilsback, K. R., Pierce, J. R., McMeeking, G. R., and Farmer, D. K.: Revisiting particle dry deposition and its role in radiative effect estimates, *Proceedings of the National Academy of Sciences*, 117, 26 076–26 082, <https://doi.org/10.1073/pnas.2014761117>, 2020.

Farmer, D. K., Boedicker, E. K., and DeBolt, H. M.: Dry Deposition of Atmospheric Aerosols: Approaches, Observations, and Mechanisms, *Annual review of physical chemistry*, 72, 375–397, <https://doi.org/10.1146/annurev-physchem-090519-034936>, 2021.

Foken, T.: Decoupling between the atmosphere and the underlying surface during stable stratification, *Boundary-Layer Meteorology*, 187, 117–140, <https://doi.org/https://doi.org/10.1007/s10546-022-00746-1>, 2023.

Foken, T. and Mauder, M.: *Micrometeorology*, Springer Atmospheric Sciences, Springer International Publishing, Cham, <https://doi.org/10.1007/978-3-031-47526-9>, 2024.

Ghahreman, R., Gong, W., Galí, M., Norman, A.-L., Beagley, S. R., Akingunola, A., Zheng, Q., Lupu, A., Lizotte, M., Levasseur, M., and Leitch, W. R.: Dimethyl sulfide and its role in aerosol formation and growth in the Arctic summer – a modelling study, *Atmos. Chem. Phys.*, 19, 14 455–14 476, <https://doi.org/10.5194/acp-19-14455-2019>, 2019.

Gong, X., Zhang, J., Croft, B., Yang, X., Frey, M. M., Bergner, N., Chang, R. Y.-W., Creamean, J. M., Kuang, C., Martin, R. V., Ranjithkumar, A., Sedlacek, A. J., Uin, J., Willmes, S., Zawadowicz, M. A., Pierce, J. R., Shupe, M. D., Schmale, J., and Wang, J.: Arctic warming by abundant fine sea salt aerosols from blowing snow, *Nature geoscience*, 16, 768–774, <https://doi.org/10.1038/s41561-023-01254-8>, 2023.

Gormley, P. G. and Kennedy, M.: Diffusion from a Stream Flowing through a Cylindrical Tube, *Proc. Royal Irish Academy*, pp. 163–169, 1949.

Gourdal, M., Lizotte, M., Massé, G., Gosselin, M., Poulin, M., Scarratt, M., Charette, J., and Levasseur, M.: Dimethyl sulfide dynamics in first-year sea ice melt ponds in the Canadian Arctic Archipelago, *Biogeosciences*, 15, 3169–3188, <https://doi.org/10.5194/bg-15-3169-2018>, 2018.

Grönlund, A., Nilsson, D., Koponen, I. K., Virkkula, A., and Hansson, M. E.: Aerosol dry deposition measured with eddy-covariance technique at Wasa and Aboa, DronningMaud Land, Antarctica, *Annals of Glaciology*, 35, 355–361, <https://doi.org/10.3189/172756402781816519>, 2002.

Guy, H., Murto, S., Karalis, M., and Tjernström, M.: Ice station micrometeorological data from expedition ARTofMELT, Arctic Ocean, 2023. Dataset version 1, Bolin Centre Database., <https://doi.org/10.17043/oden-artofmelt-2023-micrometeorology-ice-station-1>, 2024.

Hall, A.: The Role of Surface Albedo Feedback in Climate, *Journal of Climate*, 17, 1550–1568,

Held, A., Brooks, I. M., Leck, C., and Tjernström, M.: On the potential contribution of open lead particle emissions to the central Arctic aerosol concentration, *Atmos. Chem. Phys.*, 11, 3093–3105, <https://doi.org/10.5194/acp-11-3093-2011>, 2011a.

Held, A., Orsini, D. A., Vaattovaara, P., Tjernström, M., and Leck, C.: Near-surface profiles of aerosol number concentration and temperature over the Arctic Ocean, *Atmospheric Measurement Techniques*, 4, 1603–1616, <https://doi.org/10.5194/amt-4-1603-2011>, 2011b.

Holton, J. R. and Hakim, G. J.: An introduction to dynamic meteorology, Academic Press, Amsterdam, fifth edition edn., 2013.

Kerminen, V.-M. and Leck, C.: Sulfur chemistry over the central Arctic Ocean during the summer: Gas-to-particle transformation, *Journal of Geophysical Research: Oceans*, 106, 32 087–32 099, <https://doi.org/10.1029/2000JD900604>, 2001.

Kulkarni, P., Baron, P. A., and Willeke, K., eds.: *Aerosol Measurement: Principles, Techniques, and Applications*, Wiley, J, New York, NY, 3., auflage edn., 2011.

Lapere, R., Marelle, L., Rampal, P., Brodeau, L., Melsheimer, C., Spreen, G., and Thomas, J. L.: Modeling the contribution of leads to sea spray aerosol in the high Arctic, *Atmos. Chem. Phys.*, <https://doi.org/https://doi.org/10.5194/acp-24-12107-2024>, 2024.

Lawler, M. J., Saltzman, E. S., Karlsson, L., Zieger, P., Salter, M., Baccarini, A., Schmale, J., and Leck, C.: New Insights Into the Composition and Origins of Ultrafine Aerosol in the Summertime High Arctic, *Geophysical Research Letters*, 48, <https://doi.org/10.1029/2021GL094395>, 2021.

Li, X., Krueger, S. K., Strong, C., Mace, G. G., and Benson, S.: Midwinter Arctic leads form and dissipate low clouds, *Nature Communications*, 11, 206, <https://doi.org/https://doi.org/10.1038/s41467-019-14074-5>, 2020.

Liu, C., Yang, Q., Gao, Z., Shupe, M. D., Han, B., Zhang, H., Peng, S., Xi, X., and Chen, D.: The Role of Non-Local Effects on Surface Sensible Heat Flux Under Different Types of Thermal Structures Over the Arctic Sea-Ice Surface, *Geophysical Research Letters*, 51, <https://doi.org/10.1029/2023GL106753>, 2024.

55 Lüers, J. and Bareiss, J.: The effect of misleading surface temperature estimations on the sensible heat fluxes at a high Arctic site – the Arctic Turbulence Experiment 2006 on Svalbard (ARCTEX-2006), *Atmos. Chem. Phys.*, pp. 157–168, <https://doi.org/10.5194/acp-10-157-2010>, 2010.

Lüers, J. and Bareiss, J.: Direct near-surface measurements of sensible heat fluxes in the Arctic tundra applying eddy covariance and laser scintillometry—the Arctic Turbulence Experiment 2006 on Svalbard (ARCTEX-2006), *Theor. Appl. Climatol.*, pp. 387–402, <https://doi.org/10.1007/s00704-011-0400-5>, 2011.

Markuszewski, P., Nilsson, E. D., Zinke, J., Mårtensson, E. M., Salter, M., Makuch, P., Kitowska, M., Niedźwiecka-Wróbel, I., Drozdowska, V., Lis, D., et al.: Multi-year gradient measurements of sea spray fluxes over the Baltic Sea and the North Atlantic Ocean, *Atmospheric Chemistry and Physics*, 24, 11 227–11 253, <https://doi.org/10.5194/acp-24-11227-2024>, 2024.

Mauritsen, T., Sedlar, J., Tjernström, M., Leck, C., Martin, M., Shupe, M., Sjogren, S., Sierau, B., Persson, P. O. G., Brooks, I. M., and Swietlicki, E.: An Arctic CCN-limited cloud-aerosol regime, *Atmos. Chem. Phys.*, 11, 165–173, <https://doi.org/10.5194/acp-11-165-2011>, 2011.

Meredith, M., Sommerkorn, M., Cassotta, S., Derksen, C., Ekyakin, A., Hollowed, A., Kofinas, G., Mackintosh, A., Melbourne-Thomas, J., Muelbert, M., Ottersen, G., Pritchard, H., and Schuur, E.: Polar Regions: In: *IPCC Special Report on the Ocean and Cryosphere in a Changing Climate*, IPCC, [https://www.ipcc.ch/site/assets/uploads/sites/3/2019/11/07\\_SROCC\\_Ch03\\_FINAL.pdf](https://www.ipcc.ch/site/assets/uploads/sites/3/2019/11/07_SROCC_Ch03_FINAL.pdf), 2019.

Moschos, V., Dzepina, K., Bhattu, D., Lamkaddam, H., Casotto, R., Daellenbach, K. R., Canonaco, F., Rai, P., Aas, W., Becagli, S., Calzolai, G., Eleftheriadis, K., Moffett, C. E., Schnelle-Kreis, J., Severi, M., Sharma, S., Skov, H., Vestenius, M., Zhang, W., Hakola, H., Hellén, H., Huang, L., Jaffrezo, J.-L., Massling, A., Nøjgaard, J. K., Petäjä, T., Popovicheva, O., Sheesley, R. J., Traversi, R., Yttri, K. E., Schmale, J., Prévôt, A. S. H., Baltensperger, U., and El Haddad, I.: Equal abundance of summertime natural and wintertime anthropogenic Arctic organic aerosols, *Nature Geoscience*, 15, 196–202, <https://doi.org/10.1038/s41561-021-00891-1>, 2022.

Murto, S. and Tjernström, M.: Backward trajectories along the ship track using the LAGRANTO model, for the expedition ARTofMELT, Arctic Ocean, 2023, Bolin Centre Database., <https://doi.org/https://doi.org/10.17043/oden-artofmelt-2023-trajectories-backward-ship-1-2024>.

Murto, S., Tjernström, M., Karalis, M., and Prytherch, J.: Wind, temperature, relative humidity, surface temperature and radiation from expedition ARTofMELT, Arctic Ocean, 2023. Dataset version 1, Bolin Centre Database., <https://doi.org/10.17043/oden-artofmelt-2023-micromet-oden-1>, 2024.

Nilsson, E., Rannik, Ü., Swietlicki, E., Leck, C., Aalto, P. P., Zhou, J., and Norman, M.: Turbulent aerosol fluxes over the Arctic Ocean: 2. Wind-driven sources from the sea, *Journal of Geophysical Research: Atmospheres*, 106, 32 139–32 154, <https://doi.org/https://doi.org/10.1029/2000JD900747>, 2001.

Nilsson, E. D. and Rannik, Ü.: Turbulent aerosol fluxes over the Arctic Ocean: 1. Dry deposition over sea and pack ice, *Journal of Geophysical Research: Oceans*, 106, 32 125–32 137, <https://doi.org/10.1029/2000JD900605>, 2001.

Norris, S. J., Brooks, I. M., Leeuw, G., Sirevaag, A., Leck, C., Brooks, B. J., Birch, C. E., and Tjernstrom, M.: Measurements of bubble size spectra within leads in the Arctic summer pack ice, *Ocean Science*, <https://doi.org/10.5194/osd-7-1739-2010>, 2011.

Park, K., Kim, I., Choi, J.-O., Lee, Y., Jung, J., Ha, S.-Y., Kim, J.-H., and Zhang, M.: Unexpectedly high dimethyl sulfide concentration in high-latitude Arctic sea ice melt ponds, *Environmental science. Processes & impacts*, 21, 1642–1649, <https://doi.org/10.1039/c9em00195f>, 2019.

Peltola, O., Lapo, K., and Thomas, C.: A physics-based universal indicator for vertical decoupling and mixing across canopies architectures and dynamic stabilities, *Geophysical Research Letters*, 48, e2020GL091 615, <https://doi.org/https://doi.org/10.1029/2020GL091615>, 2021.

Persson, P. O. G., Fairall, C. W., Andreas, E. L., Guest, P. S., and Perovich, D. K.: Measurements near the Atmospheric Surface Flux Group tower at SHEBA: Near-surface conditions and surface energy budget, *Journal of Geophysical Research: Oceans*, 107, <https://doi.org/10.1029/2000JC000705>, 2002.

Petelski, T.: Marine aerosol fluxes over open sea calculated from vertical concentration gradients, *Journal of aerosol science*, 34, 359–371, [https://doi.org/https://doi.org/10.1016/S0021-8502\(02\)00191-9](https://doi.org/https://doi.org/10.1016/S0021-8502(02)00191-9), 2003.

Petelski, T. and Piskozub, J.: Vertical coarse aerosol fluxes in the atmospheric surface layer over the North Polar Waters of the Atlantic, *Journal of Geophysical Research: Oceans*, 111, <https://doi.org/https://doi.org/10.1029/2005JC003295>, 2006.

Petelski, T., Piskozub, J., and Paplińska-Swerpel, B.: Sea spray emission from the surface of the open Baltic Sea, *Journal of Geophysical Research: Oceans*, 110, <https://doi.org/https://doi.org/10.1029/2004JC002800>, 2005.

Prytherch, J., Yelland, M. J., Brooks, I. M., Tupman, D. J., Pascal, R. W., Moat, B. I., and Norris, S. J.: Motion-correlated flow distortion and wave-induced biases in air-sea flux measurements from ships, *Atmospheric Chemistry and Physics*, 15, 10 619–10 629, <https://doi.org/https://doi.org/10.5194/acp-15-10619-2015>, 2015.

Prytherch, J., Brooks, I., Guy, H., Karalis, M., Murto, S., and Tjernström, M.: Micrometeorological data from icebreaker Oden's foremast during expedition ARTofMELT, Arctic Ocean, 2023. Dataset version 1., Bolin Centre Database., <https://doi.org/10.17043/oden-artofmelt-2023-micromet-oden-1>, 2024.

Rantanen, M., Karpechko, A. Y., Lipponen, A., Nordling, K., Hyvärinen, O., Ruosteenoja, K., Vihma, T., and Laaksonen, A.: The Arctic has warmed nearly four times faster than the globe since 1979, *Communications Earth & Environment*, 3, <https://doi.org/10.1038/s43247-022-00498-3>, 2022.

Saylor, R. D., Baker, B. D., Lee, P., Tong, D., Pan, L., and Hicks, B. B.: The particle dry deposition component of total deposition from air quality models: right, wrong or uncertain?, *Tellus B: Chemical and Physical Meteorology*, 71, 1550 324, <https://doi.org/10.1080/16000889.2018.1550324>, 2022.

Schmale, J., Zieger, P., and Ekman, A. M. L.: Aerosols in current and future Arctic climate, *Nature Climate Change*, 11, 95–105, <https://doi.org/10.1038/s41558-020-00969-5>, 2021.

Scott, W. D. and Levin, Z.: Open channels in sea ice (leads) as ion sources, *Science (New York, N.Y.)*, 177, 425–426, <https://doi.org/10.1126/science.177.4047.425>, 1972.

Secretariat, S. P. R.: ARTofMELT 2023, <https://www.polar.se/en/expeditions/previous-expeditions/arctic/artofmelt-2023/>, 2024.

Sedlar, J., Tjernström, M., Mauritsen, T., Shupe, M. D., Brooks, I. M., Persson, P. O. G., Birch, C. E., Leck, C., Sirevaag, A., and Nicolaus, M.: A transitioning Arctic surface energy budget: the impacts of solar zenith angle, surface albedo and cloud radiative forcing, *Climate Dynamics*, 37, 1643–1660, <https://doi.org/10.1007/s00382-010-0937-5>, num Pages: 18, 2011.

Stevens, R. G., Loewe, K., Dearden, C., Dimitrelos, A., Possner, A., Eirund, G. K., Raatikainen, T., Hill, A. A., Shipway, B. J., Wilkinson, J., Romakkaniemi, S., Tonttila, J., Laaksonen, A., Korhonen, H., Connolly, P., Lohmann, U., Hoose, C., Ekman, A. M. L., Carslaw, K. S., and Field, P. R.: A model intercomparison of CCN-limited tenuous clouds in the high Arctic, *Atmospheric Chemistry and Physics*, 18, 11 041–11 071, <https://doi.org/https://doi.org/10.5194/acp-18-11041-2018>, 2018.

Tjernström, M. and Zieger, P.: Expedition report Atmospheric rivers and the onset of Arctic melt, ARTofMELT, 2023 with icebreaker Oden, Swedish Polar Research Secretariat, <https://www.polar.se/en/expeditions/expedition-reports>, 2025.

Tremblay, S., Picard, J.-C., Bachelder, J. O., Lutsch, E., Strong, K., Fogal, P., Leaitch, W. R., Sharma, S., Kolonjari, F., Cox, C. J., Chang, R. Y.-W., and Hayes, P. L.: Characterization of aerosol growth events over Ellesmere Island during the summers of 2015 and 2016, *Atmospheric Chemistry and Physics*, 19, 5589–5604, <https://doi.org/10.5194/acp-19-5589-2019>, 2019.

Twomey, S.: The Influence of Pollution on the Shortwave Albedo of Clouds, *Journal of the Atmospheric Sciences*, 34, 1149–1152, 1977.

Von Der Weiden, S.-L., Drewnick, F., and Borrmann, S.: Particle Loss Calculator – a new software tool for the assessment of the performance of aerosol inlet systems, *Atmospheric Measurement Techniques*, 2, 479–494, <https://doi.org/10.5194/amt-2-479-2009>, 2009.

Watanabe, T. K. and Pfeiffer, M.: A simple Monte Carlo approach to estimate the uncertainties of SST and  $\delta^{18}\text{O}_{\text{SW}}$  inferred from coral proxies, *Geochemistry, Geophysics, Geosystems*, 23, e2021GC009813, <https://doi.org/10.1029/2021GC009813>, 2022.

Wendisch, M., Macke, A., Ehrlich, A., Lüpkes, C., Mech, M., Chechin, D., Dethloff, K., Velasco, C. B., Bozem, H., Brückner, M., Clemen, H.-C., Crewell, S., Donth, T., Dupuy, R., Ebelt, K., Egerer, U., Engelmann, R., Engler, C., Eppers, O., Gehrman, M., Gong, X., 635 Gottschalk, M., Gourbeyre, C., Griesche, H., Hartmann, J., Hartmann, M., Heinold, B., Herber, A., Herrmann, H., Heygster, G., Hoor, P., Jafariserajehlou, S., Jäkel, E., Järvinen, E., Jourdan, O., Kästner, U., Kecorius, S., Knudsen, E. M., Köllner, F., Kretzschmar, J., Lelli, L., Leroy, D., Maturilli, M., Mei, L., Mertes, S., Mioche, G., Neuber, R., Nicolaus, M., Nomokonova, T., Notholt, J., Palm, M., van Pinxteren, M., Quaas, J., Richter, P., Ruiz-Donoso, E., Schäfer, M., Schmieder, K., Schnaiter, M., Schneider, J., Schwarzenböck, A., Seifert, P., Shupe, M. D., Siebert, H., Spreen, G., Stapf, J., Stratmann, F., Vogl, T., Welti, A., Wex, H., Wiedensohler, A., Zanatta, M., and Zeppenfeld, 640 S.: The Arctic Cloud Puzzle: Using ACLOUD/PASCAL Multiplatform Observations to Unravel the Role of Clouds and Aerosol Particles in Arctic Amplification, *Bulletin of the American Meteorological Society*, 100, 841–871, <https://doi.org/10.1175/BAMS-D-18-0072.1>, 2019.

Wendisch, M., Brückner, M., Crewell, S., Ehrlich, A., Notholt, J., Lüpkes, C., Macke, A., Burrows, J. P., Rinke, A., Quaas, J., Maturilli, 645 M., Schemann, V., Shupe, M. D., Akansu, E. F., Barrientos-Velasco, C., Bärfuss, K., Blechschmidt, A.-M., Block, K., Bougoudis, I., Bozem, H., Böckmann, C., Bracher, A., Bresson, H., Bretschneider, L., Buschmann, M., Chechin, D. G., Chylik, J., Dahlke, S., Deneke, H., Dethloff, K., Donth, T., Dorn, W., Dupuy, R., Ebelt, K., Egerer, U., Engelmann, R., Eppers, O., Gerdes, R., Gierens, R., Gorodetskaya, I. V., Gottschalk, M., Griesche, H., Gryanik, V. M., Handorf, D., Harm-Altstädtter, B., Hartmann, J., Hartmann, M., Heinold, B., Herber, A., Herrmann, H., Heygster, G., Höschel, I., Hofmann, Z., Hölemann, J., Hünerbein, A., Jafariserajehlou, S., Jäkel, E., Jacobi, C., Janout, M., Jansen, F., Jourdan, O., Jurányi, Z., Kalesse-Los, H., Kanzow, T., Käthner, R., Kliesch, L. L., Klingebiel, M., Knudsen, E. M., Kovács, 650 T., Körtke, W., Krampe, D., Kretzschmar, J., Kreyling, D., Kulla, B., Kunkel, D., Lampert, A., Lauer, M., Lelli, L., von Lerber, A., Linker, O., Löhner, U., Lonardi, M., Losa, S. N., Losch, M., Maahn, M., Mech, M., Mei, L., Mertes, S., Metzner, E., Mewes, D., Michaelis, J., Mioche, G., Moser, M., Nakoudi, K., Neggers, R., Neuber, R., Nomokonova, T., Oelker, J., Papakonstantinou-Presvelou, I., Pätzold, F., Pefanis, V., Pohl, C., van Pinxteren, M., Radovan, A., Rhein, M., Rex, M., Richter, A., Risse, N., Ritter, C., Rostosky, P., Rozanov, V. V., Donoso, E. R., Saavedra Garfias, P., Salzmann, M., Schacht, J., Schäfer, M., Schneider, J., Schnierstein, N., Seifert, P., Seo, S., Siebert, 655 H., Soppa, M. A., Spreen, G., Stachlewska, I. S., Stapf, J., Stratmann, F., Tegen, I., Viceto, C., Voigt, C., Vountas, M., Walbröl, A., Walter, M., Wehner, B., Wex, H., Willmes, S., Zanatta, M., and Zeppenfeld, S.: Atmospheric and Surface Processes, and Feedback Mechanisms Determining Arctic Amplification: A Review of First Results and Prospects of the (AC)3 Project, *Bulletin of the American Meteorological Society*, 104, E208–E242, <https://doi.org/10.1175/bams-d-21-0218.1>, 2023.

660 Wieringa, J.: A revaluation of the Kansas mast influence on measurements of stress and cup anemometer overspeeding, *Boundary layer meteorology*, 18, 411–430, <https://doi.org/10.1007/BF00119497>, 1980.

Willis, M. D., Leaitch, W. R., and Abbatt, J. P.: Processes Controlling the Composition and Abundance of Arctic Aerosol, *Reviews of Geophysics*, 56, 621–671, <https://doi.org/10.1029/2018RG000602>, 2018.

Yang, X., Pyle, J. A., and Cox, R. A.: Sea salt aerosol production and bromine release: Role of snow on sea ice, *Geophysical Research Letters*, 35, <https://doi.org/10.1029/2008GL034536>, 2008.

665 Yang, X., Frey, M. M., Rhodes, R. H., Norris, S. J., Brooks, I. M., Anderson, P. S., Nishimura, K., Jones, A. E., and Wolff, E. W.: Sea salt aerosol production via sublimating wind-blown saline snow particles over sea ice: parameterizations and relevant microphysical mechanisms, *Atmospheric Chemistry and Physics*, 19, 8407–8424, <https://doi.org/10.5194/acp-19-8407-2019>, 2019.