# Peer review of "Particle flux-gradient relationships in the high Arctic: Emission and deposition patterns across three surface types"

_EGUsphere, 2025_

## Author Comment (AC1)

**General comment**

The paper reports measurements of particle fluxes using a gradient approach in Arctic to investigate emission and deposition over different surfaces. Measurements were done during the ARTofMEL expedition in an environment that is difficult to characterise in terms of particle fluxes being challenging for the measurement setup. I believe that the results are interesting and may be of interest for the scientific community. There are a few aspects that should be improved as detailed in my specific comments.

We sincerely thank the reviewer for his/her time, effort and thoughtful feedback. The constructive comments have been highly helpful in improving the manuscript. Below, we provide a point-by-point response to each of the reviewer's comments. Reviewer comments are presented in black, our response in blue, and changes in the manuscript in orange.

**Specific comments**

It is used to mention the normalised flux that is essentially what is indicated in other studies as deposition velocity. Why not using the more common deposition velocity?

We agree with the referee that the formulation deposition velocity is more common. Deposition velocity is usually used to refer to true concentration-dependent dry deposition. As stated in equation 11, $V_D$ is defined as a normalized flux.

The relationship between the flux and mean concentration is different for a surface emission flux and a deposition flux. For deposition, the flux is dependent on the mean particle concentration and the strength of turbulence - losses to the surface occur via turbulent impaction and Brownian diffusion. For a given size and turbulent conditions, the flux scales linearly with the particle concentration. A source from the surface is controlled by some surface processes, for example, bubble bursting processes in open water. This is independent of the concentration in the atmosphere.

Given an existing particle concentration in the atmosphere, both emission and deposition processes can operate at the same time, and the net measured flux (emission and deposition) does depend on the ambient concentration, but only through the deposition term.

Due to these reasons, to emphasize that we do not refer only to deposition processes, we would like to keep the term "normalized flux".

Line 211: To emphasize that we do not only refer to deposition processes we use the term normalized flux for $V_D$.

It seems that when it is mentioned net deposition or net emission it is referred to a single 20-minute period, it may create confusion with long-term average of fluxes.

The cases where it is not entirely clear whether a single 20-minute period or a long-term average is meant have been adjusted accordingly:

Line 329: However, taking into account the estimated maximum uncertainty (grey shading in Fig. 5), fluxes may be positive or negative but with a tendency toward net particle emission 20-minute intervals.

Line 340: Rather, both net particle emission and deposition intervals may be observed in this mixed area of influence.

Line 381: Net particle emission intervals from a closed ice surface like on 1 June (Fig. 6 (b) has also been observed in other studies, e.g. in Nilsson et al. (2001). As particle fluxes and normalized fluxes are strongly dependent on particle size (Nilsson and Rannik, 2001), a changing size distribution (Fig. 6 (e), (f)) could explain a change from net particle deposition to net particle emission intervals.

Line 432: On several occasions, net particle emission intervals occurred under the influence of closed ice.

Figure 5 / 6 / A3 / A4: The brown color indicates net particle emission 20-minute intervals, and the blue color indicates net particle deposition 20-minute intervals.

In equations (1) to (5) it is used the capital U' for fluctuations while this was not done for other velocity components, why?

To avoid confusion with the longitudinal component of wind speed u, the capital U was used for the horizontal wind speed.

Line 45. I would say ideally 10 Hz because very often it is a lower resolution, also in the EC measurement here. I also suggest to mention that in EC measurements involving particles, or more in general closed path sensors, it is important the first order time response of the inlet rather than the sampling frequency, because this is often a more limiting factor for fast instruments see for example the discussion in Conte et al (2018, Science of the Total Environment 622, 1067-1078).

We changed that sentence and added the information about the closed path sensors.

Line 45: Difficulties arise particularly from the need for particle measurements with a high temporal resolution of ideally 10 Hz, which is essential for flux measurements to capture the fine-scale fluctuations of small variations, combined with the challenge of low particle concentrations. In addition to the first order time response of the device, fluctuation dampening of the inlet must also be taken into account, which is often the limiting factor (Conte et a. 2018).

Figure 2. It would be interesting to add the comparison among the two EC systems, ice mast and ship mast to discuss is the differences are due to the different location or to the different methods (EC and gradient). The same for Figure 3. Do you have an interpretation on why the comparison for H is significantly worse than that for u*?

A more detailed discussion on why the comparison for H is worse than that for u* can be found in the comments of reviewer 2 (Thomas Foken) and our response to these comments. Briefly,

when using the gradient method, H is calculated not only from the wind profile (as for u*) but also from the temperature profile. For near-surface profiles, the lowest height close to the ice surface will exhibit very low temperatures in all situations and "may 'simulate' stable stratification", as pointed out by reviewer 2. Thus, the gradient can become very large and vertical decoupling may further affect the comparison, especially when the friction velocity is very low.

We followed the reviewer's recommendation and added the comparison among the two Eddy Covariance Systems:

Line 256: The results demonstrate that the eddy covariance measurements on the ship mast and ice mast show the highest degree of agreement (Fig.2(a)). Nevertheless, as can be seen in Fig. 2 (c), many of the $u_*$ values determined at the ice mast and gradient system sites are in good accordance and close to the 1:1 line. In particular, when both systems are influenced by the same surface type (closed ice), there is good agreement of u*. The agreement of the $u_*$ values determined at the ship mast and gradient system sites (Fig. 2 (b) is slightly lower, but still consistent with expectations. The deviation of $u_*$ between the gradient system and ship mast measurements was less than 50 % in 63 % of the cases. The deviation of u* between the gradient system and ice mast measurements was less than 50 % in 77 % of the cases, and between the ship mast and ice mast in 89 % of the cases.

[Figure]

Figure 2. (a) Scatter plot between friction velocities [u*, m s$^{-1}$] measured by the ship mast (y-axis) and the ice mast (x-axis). (b) Scatter plot between friction velocities measured by the gradient system (y-axis) and the ice mast (x-axis). (c) Scatter plot between friction velocities measured by the gradient system (y-axis) and the ship mast (x-axis). The dark blue lines symbolize the 1:1 line, the turquoise lines the 50 % deviations. The beige line shows the linear regression fit. The three different symbol colors in (b) and (c) divide the measurements according to the surface type, which influenced the gradient system.

Line 264: The comparison of the sensible heat fluxes between the three systems shows the dependence of the sensible heat flux on the type of surface surrounding the system (Fig. 3 (b), (c). As for the friction velocity, the sites ship mast and ice mast (Fig. 3 (c)), both evaluated with the eddy covariance method, show the highest degree of agreement. The comparison

between the ice mast and gradient system shows that for the narrow lead surface type, the fluxes differ in sign and thus in direction. Also, when the gradient system is influenced by wide leads, clear differences can be seen, which is to be expected due to the different surface types influencing the two systems. However, when influenced by the same surface type (closed ice) the values are of the same magnitude (Fig. 3 (c)). When comparing the gradient system with the ship mast (Fig. 3 (b)), the opposite sign is also evident for the narrow lead. For the wide lead, high positive sensible heat fluxes are observed with both systems. This is consistent with the ship mast being surrounded by wide leads at the same time as the gradient system.

[Figure]

Figure 3. (a) Scatter plot between sensible heat fluxes [H, W m$^{-2}$] measured by the ship mast (y-axis) and the ice mast (x-axis). (b) Scatter plot between sensible heat fluxes measured by the gradient system (y-axis) and ship mast (x-axis). (c) Scatter plot between sensible heat fluxes measured by the gradient system (y-axis) and ice mast (x-axis). The dark blue lines symbolize the 1:1 line. The three different colors divide the measurements according to the surface type, which influenced the gradient system.

Line 244. The uncertainties of fluxes are quite high, it would be useful a comment if this is enough to have a robust measurement.

We agree with the referee that the low particle concentrations combined with a long sample inlet and the resulting losses lead to large uncertainties. The uncertainties calculated by the Monte Carlo simulation are the maximum uncertainties, which may significantly exceed the true uncertainty. In lines 321-324 we have commented that exact quantification is subject to large uncertainties and that the focus should therefore be on tendencies towards net emission or net deposition, depending on the surface type.

Table 2. Better to write 0.03-0.04 in the first raw and 0.005 in the second because the inerval 0.05-0.05 is not clear.

We agree with the referee and have changed the values.

Figure 6. What do you mean with normalised concentrations? Why not showing the size distribution with the typical normalisation using dLog?

We agree that the term normalized concentrations might be confusing here. The size distribution data are expressed in the conventional way using dN/dlogDp, and in addition, normalized by dividing by the total integrated concentration for each scan. In this way, the relative contribution of different size classes to the overall size distribution in Figure 6 e) and f) can be compared independent of the total particle concentration. The term "Normalized dN/dlogDp" better reflects the presented data and the term has been changed accordingly in Figure 6 and the caption of Figure 6.

Figure 6. […] (f) Normalized dN/dlogDp [%] particle size distribution from 15 to 790 nm (left y-axis) with total particle number concentration, measured with a DMPS on the 4th deck of Oden (white line; right y-axis).

[Figure]

---

## Author Comment (AC2)

This publication investigates particle deposition (emission) over ice surfaces and thus an extremely important problem of changes in surface albedo and possible influences on the Arctic climate. The measurement concept corresponds to the current technical possibilities and the authors are recognised experts in this field. The measurement under Arctic conditions is a particular challenge. The theory used is state of the art, but the restriction to neutral stratification (which may not really correspond to reality after all) would not have been necessary, as the curvature of the gradients can certainly be taken into account by universal functions when determining the gradient (Foken and Mauder, 2024).

We sincerely thank the reviewer Thomas Foken for his time, effort and thoughtful feedback. The constructive comments have been highly helpful in improving the manuscript. Below, we provide a point-by-point response to each of the reviewer's comments. Reviewer comments are presented in black, our response in blue, and changes in the manuscript in orange.

The general verification of the systems by determining the friction velocity is very useful and should also be used for further classification of the measurements if necessary. The deviations shown in Fig. 2a are typical for the gradient-eddy-covariance comparison, but in Fig. 2b the measurements should be labelled with a different symbol if there are significant differences in the footprint of the two systems or if the eddy mast is located to leeward of the ship.

We follow the reviewer's recommendation and now use different symbol colors for the surface type influencing the gradient system. In addition, we add a third scatter plot comparing u* measured by the ship mast and by the ice mast to give a general impression of differences due to different footprints. The surface type for the measurements on the ship mast has not been evaluated in detail. For the entire measurement campaign, care was take to align the ship with the prevailing wind direction, so the ship mast was not located to leeward of the ship for extended periods.

Line 256: The results demonstrate that the eddy covariance measurements on the ship mast and ice mast show the highest degree of agreement (Fig.2(a)). Nevertheless, as can be seen in Fig. 2 (c), many of the $u_*$ values determined at the ice mast and gradient system sites are in good accordance and close to the 1:1 line. In particular, when both systems are influenced by the same surface type (closed ice), there is good agreement of u*. The agreement of the $u_*$ values determined at the ship mast and gradient system sites (Fig. 2 (b) is slightly lower, but still consistent with expectations. The deviation of $u_*$ between the gradient system and ship mast measurements was less than 50 % in 63 % of the cases. The deviation of u* between the gradient system and ice mast measurements was less than 50 % in 77 % of the cases, and between the ship mast and ice mast in 89 % of the cases.

[Figure]

Figure 2. (a) Scatter plot between friction velocities [u*, m s⁻¹] measured by the ship mast (y-axis) and the ice mast (x-axis). (b) Scatter plot between friction velocities measured by the gradient system (y-axis) and the ice mast (x-axis). (c) Scatter plot between friction velocities measured by the gradient system (y-axis) and the ship mast (x-axis). The dark blue lines symbolize the 1:1 line, the turquoise lines the 50 % deviations. The beige line shows the linear regression fit. The three different symbol colors in (b) and (c) divide the measurements according to the surface type, which influenced the gradient system.

The reviewer cannot follow the discussion of the results of the sensible heat flux (Fig. 3 and 4). The gradient mast does not have a uniform footprint, i.e. the lowest height has a very small footprint which is probably exclusively ice in all situations. This means that the temperature is also very low in all situations and may 'simulate' stable stratification. This can be seen very clearly with 'Narrow Lead', where the eddy-covariance measurements correctly show a positive sensible heat flux, while the gradient mast always indicates stable stratification.

We fully agree that the different measurement footprints at different heights of the gradient mast will complicate the interpretation of the profiles. Indeed, for 'Narrow Lead' conditions the scatter plots in the revised Figure 3 (b,c) clearly show that the sensible heat fluxes derived from the gradient mast are mostly negative and almost always lower than the mostly positive eddy-covariance fluxes. We add a brief discussion of the potentially non-uniform footprints in the revised manuscript.

Line 264: The comparison of the sensible heat fluxes between the three systems shows the dependence of the sensible heat flux on the type of surface surrounding the system (Fig. 3 (b), (c). As for the friction velocity, the sites ship mast and ice mast (Fig. 3 (a)), both evaluated with the eddy covariance method, show the highest degree of agreement. The comparison between the ice mast and gradient system shows that for the narrow lead surface type, the fluxes differ in sign and thus in direction. It must be noted that the footprint area affecting the lowest measurement heights of the gradient system becomes very small, and even for the narrow lead surface type, the ice surface with low temperatures will influence these measurements. Also, when the gradient system is influenced by wide leads, clear differences can be seen, which is to be expected due to the different surface types influencing the two systems. However, when influenced by the same surface type (closed ice) the values are of the same magnitude (Fig. 3 (c)). When comparing the gradient system with the ship mast (Fig. 3 (b)), the opposite sign is also evident for the narrow lead. For the wide lead, high positive

sensible heat fluxes are observed with both systems. This is consistent with the ship mast being surrounded by wide leads at the same time as the gradient system.

[Figure]

Figure 3. (a) Scatter plot between sensible heat fluxes [H, W m$^{-2}$] measured by the ship mast (y-axis) and the ice mast (x-axis). (b) Scatter plot between sensible heat fluxes measured by the gradient system (y-axis) and ship mast (x-axis). (c) Scatter plot between sensible heat fluxes measured by the gradient system (y-axis) and ice mast (x-axis). The dark blue lines symbolize the 1:1 line. The three different colors divide the measurements according to the surface type, which influenced the gradient system.

The situation becomes even more problematic with 'Closed Ice'. The measurements are increasingly stable and can in no way be assigned to the neutral range (however, the specified range for z/L is also very narrowly defined). The gradient mast in particular measures relatively large downward sensible heat fluxes, i.e. the gradient is comparatively large. Various phenomena such as decoupling, counter-gradients and coherent structures occur particularly at very low friction velocities (Foken, 2023;de La Casinière, 1974;Grachev et al., 2005;Sodemann and Foken, 2005;Lüers and Bareiss, 2010). To discuss the data, they should be categorised into u* classes. With regard to the interpretation of the particle fluxes, fluxes with u*<0.10... 0.15 m/s must probably be excluded after this investigation. The discussion of all the phenomena mentioned is too complicated and the data set only allows this in part. It may be possible to estimate the possibility of decoupling with the Brunt-Väisälä frequency (Foken, 2023;Peltola et al., 2021).

We agree with the referee that especially with the surface type 'Closed ice' stable conditions close to the ground can exist. We thank the reviewer for suggesting to categorize our data by u* for the discussion, and to estimate possible decoupling with the Brunt-Väisälä frequency. First, we calculated the decoupling metric Ω according to Peltola et al. (2021) for flat surfaces without emergent vegetation. The height and temperature differences were calculated from the gradient profiles and thus served as a basis for N. σw was taken from the ice mast data. Peltola et al. (2021) define the following three regimes for the decoupling metric:

Ω ≥ 0.61 -> coupled

0.43 ≤ Ω < 0.61 -> weakly coupled

Ω < 0.43 -> decoupled

As suspected, the possibility of decoupling occurs frequently. Figure R1 shows that decoupling was likely particularly when the lift was surrounded by closed ice (29.05.-05.06.).

[Figure]

Figure R1: Decoupling metric Ω, calculated with the Brunt-Väisälä frequency (Peltola et al. 2021).

Second, based on these findings, we followed the reviewer's recommendation and categorized all flux data in two u* classes. All means and medians of the turbulent particle and sensible heat fluxes as well as the normalised fluxes in the revised manuscript now refer to periods when u* ≥ 0.15 m s$^{-1}$. The number of intervals has also been adjusted accordingly. A distinction has been made in the figures between intervals above and below the u* threshold.

We agree with the referee that it is out of the scope of this manuscript to discuss all the mentioned phenomena in detail. Here, we have made the following changes in the revised manuscript:

Line 201: In the case of stable stratification, strong temperature gradients can occur very close to the surface. In order to determine periods with possible vertical decoupling, a decoupling metric based on the Brunt-Väisälä frequency has been calculated (Foken 2023, Peltola et al. 2021). While the decoupling metric indicates coupled or weakly coupled periods when

measuring at the lead, decoupling is frequently possible when measuring over closed ice surfaces. In order to exclude periods with weak turbulence and thus a high probability of vertical decoupling, the fluxes calculated from the gradient system were classified according to u*. For further discussion of the turbulent particle and sensible heat fluxes according to the flux-profile relationships, all mean and median values as well as the number of intervals are based on periods when u* ≥ 0.15 m s$^{-1}$.

Line 274: All measured values here are for conditions of u* ≥ 0.15 m s$^{-1}$.

Line 284: For this period, the low u* (beige circles) indicates possible vertical decoupling.

[Figure]

Figure 4. Sensible heat flux [H, W m$^{-2}$] on (a) 17 May and (b) 18 May 2023 from the gradient system setup influenced by wide leads with a time resolution of 20-minutes as well as sensible heat flux from the ship mast and ice mast with a time resolution of 20-minutes. (c) Sensible heat flux from the second ice camp from 29 June to 10 June 2023 from the three different measurement setups (gradient system, ice mast, and ship mast). In (a) and (b), all u* ≥ 0.15 m s$^{-1}$; in (c), gradient system data are shown in different colors depending on u*.

The conditions of the sensible heat flux naturally influence the particle gradient in the same way. At the very least, the proposed classification should be adopted. In a further study, it might be useful to investigate whether particles accumulate in the shallow layer above the ice in the event of decoupling. The layer is probably emptied of particles again with a short-term emission event. Perhaps Fig. A5 should be included in the text and compared with Fig. 3.

We agree with the referee that data analysis with respect to decoupling offers many possibilities for further analysis, especially since the data base in the Arctic for such shallow inversion layers is very limited, and we thank for this interesting suggestion. In this manuscript, we have adopted the u* classification as proposed and include a brief discussion of possible vertical decoupling. In the revised manuscript, we now distinguish particle fluxes in periods u* < 0.15 m/s and u* ≥ 0.15 m. Because Fig. A5 only serves to compare particle fluxes and wind speed conditions, we prefer to not include it in the main text. Further revisions have been indicated in our response to the previous comment.

In the conclusions, one would have to answer the question of why the ice surface is a sink for particles. Is the cause the surface itself or the stable stratification predominantly found there? Perhaps it is possible to subdivide the results into 2-3 stability classes (z/L) based on the eddy covariance data.

The z/L values based on the eddy covariance data show only neutral and stable conditions for the wide and narrow lead measurements. For the closed ice measurements, there were stable conditions temporarily, but no dependence of z/L on the particle flux direction direction is evident.

As is often the case with experimental studies, there are more questions at the end than were solved by the experiment. Thus, the manuscript should only be revised very carefully to the extent absolutely necessary, but problems should be pointed out. Possibly the discussion of the questions raised should be dealt with in another article.

Again, we sincerely thank the reviewer for his time and effort. His thoughtful feedback helped to make important revisions and substantially improve the manuscript.

Minor comments:

Line 148ff: Normalised size distribution should be defined or explained like all other normalisations.

The size distribution data are expressed in the conventional way using dN/dlogDp, and in addition, normalized by dividing by the total integrated concentration for each scan. In this way, the relative contribution of different size classes to the overall size distribution in Figure 6 e) and f) can be compared independent of the total particle concentration. The term "Normalized dN/dlogDp" better reflects the presented data and the term has been changed accordingly in Figure 6 and the caption of Figure 6.

Line 490: Please replace Foken (2017) with Foken and Mauder (2024)

The reference was replaced as suggested.

References

de La Casinière, A. C.: Heat Exchange over a Melting Snow Surface, J. Glaciol., 13, 55-72, doi: 10.3189/S0022143000023376, 1974.

Foken, T.: Decoupling between the atmosphere and the underlying surface during stable stratification, Boundary-Layer Meteorol., 187, 117-140, doi: 10.1007/s10546-022-00746-1, 2023.

Foken, T., and Mauder, M.: Micrometeorology, 3 ed., Springer, Cham, XXI, 410 pp., doi: 10.1007/978-3-031-47526-9, 2024.

Grachev, A. A., Fairall, C. W., Persson, P. O. G., Andreas, E. L., and Guest, P. S.: Stable Boundary-Layer Scaling Regimes: The Sheba Data, Boundary-Layer Meteorol., 116, 201-235, doi: 10.1007/s10546-004-2729-0, 2005.

Lüers, J., and Bareiss, J.: The effect of misleading surface temperature estimations on the sensible heat fluxes at a high Arctic site – the Arctic Turbulence Experiment 2006 on Svalbard (ARCTEX-2006), Atmospheric Chemistry and Physics, 10, 157-168, doi: 10.5194/acp-10-157-2010, 2010.

Peltola, O., Lapo, K., and Thomas, C. K.: A physics-based universal indicator for vertical decoupling and mixing across canopies architectures and dynamic stabilities, Geophys. Res. Letters, 48, e2020GL091615, doi: https://doi.org/10.1029/2020GL091615, 2021.

Sodemann, H., and Foken, T.: Special characteristics of the temperature structure near the surface, Theor. Appl. Climat., 80, 81-89, doi: 10.1007/s00704-004-0092-1, 2005.

---

## Author Comment (AC3)

This manuscript presents a well-executed field study investigating near-surface aerosol particle fluxes and sensible heat fluxes over different Arctic surface types using a novel gradient-based measurement system during the ARTofMELT expedition. The comparison with eddy covariance data enhances the credibility of the measurements. The study provides valuable insight into particle source/sink behavior over wide leads, narrow leads, and closed ice, contributing to our understanding of aerosol-cloud-sea-ice interactions in the central Arctic.

The paper is timely, methodologically sound, and clearly structured, but there are several areas where the manuscript could be improved for clarity, completeness, and scientific robustness.

We sincerely thank Piotr Markuszewski and Monica Mårtensson for their time, effort and constructive feedback. Below, we provide a response to their comments which helped in improving the manuscript. Comments are presented in black, our response in blue, and changes in the manuscript in orange.

The manuscript would benefit from a significantly broader and more critical engagement with the prior literature on gradient-based flux measurements, particularly over marine and ice surfaces. While the authors cite several key studies related to Arctic fluxes (e.g., Nilsson et al. 2001, Held et al. 2011a,b), the discussion omits earlier foundational work using gradient methods to estimate aerosol and heat fluxes in polar and marine environments.

For instance, the first gradient-based aerosol flux measurements conducted by Petelski (2003), Petelski et al. (2005), and further refined in Petelski and Piskozub (2006), including the later comment given by Andreas (2007), are highly relevant and should be referenced. Additionally, Savelyev et al. (2014) addressed fluxes under low-turbulence regimes using similar profile techniques, which is particularly relevant for the stable and weakly turbulent conditions encountered over closed ice. Authors may also find recent publication dedicated to the gradient method (Markuszewski et al., 2024).

In its current form, the manuscript gives the impression that the gradient method is underexplored in this field, which is not accurate. A proper contextualization would strengthen the justification for the study, allow a more meaningful comparison of uncertainties and assumptions, and acknowledge the methodological evolution within the field of flux-gradient applications.

We appreciate the helpful comments and suggestions, especially the numerous references. Although we focus on near-surface gradients on an ice flow in the high Arctic and it is beyond the scope of our manuscript to give a full discussion of the foundational work of gradient methods, these references can certainly help to avoid the impression that the gradient method is generally underexplored, and improve our assessment of the application of the gradient method in the high Arctic. The suggested literature has been incorporated as follows:

Line 46: An alternative method for measuring fluxes in challenging environmental conditions, such as the sea ice, is the gradient method, which is based on the theory of flux-profile relationships (e.g. Farmer et al., 2021). The gradient method has been applied in different areas. On moving platforms like ships, the gradient method was applied e.g. by Petelski (2003), Petelski and Piskozub (2006) and Petelski et al. (2005). Multi-year measurements using the gradient method have been performed, for example, by Markuszewski et al. (2024).

Markuszewski, P., Nilsson, E. D., Zinke, J., Mårtensson, E. M., Salter, M., Makuch, P., and Piskozub, J.: Multi-year gradient measurements of sea spray fluxes over the Baltic Sea and the North Atlantic Ocean, Atmos. Chem. Phys., 24, 11227–11253, https://doi.org/10.5194/acp-24-11227-2024, 2024.

Petelski, T.: Marine aerosol fluxes over open sea calculated from vertical concentration gradients, J. Aerosol Sci., 34, 359–371, https://doi.org/10.1016/S0021-8502(02)00191-9, 2003.

Petelski, T., and Piskozub, J.: Vertical coarse aerosol fluxes in the atmospheric surface layer over the North Polar Waters of the Atlantic, J. Geophys. Res.-Oceans, 111, C06039, https://doi.org/10.1029/2005JC003295, 2006.

Petelski, T., Piskozub, J., and Paplińska-Swerpel, B.: Sea spray emission from the surface of the open Baltic Sea, J. Geophys. Res.-Oceans, 110, C10023, https://doi.org/10.1029/2004JC002800, 2005.

The manuscript lacks of presentation of the raw or processed aerosol concentration profiles that underpin the flux-gradient calculations. Since the fluxes are derived from linear regressions across vertical gradients of particle number concentrations, it is essential to show representative examples of these vertical profiles to assess the validity of the method. Including a few example profiles—either in the main text or as supplementary material—would serve as a critical demonstration that the gradient system performs as intended. For instance, plots showing aerosol number concentrations at each height, along with fitted regression lines and $R^2$ values, would give readers confidence in the robustness of the derived fluxes. It is also unclear what the range of correlation coefficients (e.g., $R^2$ of the linear fit) was across the dataset, or how often profiles were rejected due to poor fits. Providing a histogram or table of regression diagnostics (e.g., slope, $R^2$, residuals) would help clarify the quality and reliability of the profiles used in flux calculations. Furthermore, it would be useful to include a discussion of profile curvature, measurement noise, or transient concentration spikes, and how these were addressed during preprocessing and averaging. Without this level of transparency, the central assumption—that vertical particle concentration gradients are well-defined and resolved—is insufficiently supported.

We follow the suggestion and now provide an example profile in the Appendix to allow the reader to follow in more detail how gradients and fluxes were derived from observed

profiles. $R^2$ values were calculated for linear regression fits of the scalar as measured and the logarithm of the measurement height, thus evaluating the deviation of the profile curvature from an ideal logarithmic profile. Regarding transient concentration spikes, raw particle concentrations values were removed before averaging if values were outside 1.5 times the interquartile range of a moving 20-minute window, as stated in section 2.2. Regarding measurement noise, we did not take into account estimates of the uncertainty of individual 1 s particle number concentration values, however, the standard deviation of 1 s particle number concentration values in the averaging periods of the profile height levels ranging from 3 % to 21 % was included as an estimate of the variability of the particle number concentration measurements when estimating the flux uncertainty using Monte Carlo simulations. This estimate includes both measurement noise and the true variation of the particle number concentration and is therefore likely an overestimation.

Line 190: Figure AX shows an example of measured particle concentration and wind speed profiles and the linear regression fits.

[Figure]

Figure AX. Example of vertical profiles of particle concentration (a) and wind speed (b) over closed ice on 5 June 2023. The beige line shows the linear regression fit with $R^2 = 0.98$ for the particle concentration gradient and $R^2 = 0.99$ for the wind speed gradient. Utilising the calculated u* = 0.15 m/s, a flux of 0.47 $10^6$ m$^{-2}$s$^{-1}$ was derived.

Another methodological weakness stems from the limited footprint characterization and the difference between systems used for intercomparison. Yet, the paper compares their fluxes as though they were co-located. While some differences are acknowledged qualitatively, there is no attempt to assess or estimate the footprints, nor to indicate whether the observed discrepancies fall within expected spatial variability. The authors should include either a footprint model (even a simplified 1D footprint estimate based on surface roughness and stability) or a discussion of fetch dependence to justify the comparability of the datasets. This is particularly important for interpreting disagreements in flux direction and magnitude over mixed or narrow lead surfaces.

Indeed, measurement footprints are only discussed qualitatively and not estimated using footprint models. However, we do not compare fluxes of the gradient system, the ice mast and the ship as though they were co-located. In the original manuscript, we clearly point out that differences between the systems are expected when the systems are influenced by different surface types, e.g. in lines 268 – 270: "… clear differences can be seen, which is to be expected due to the different surface types influencing the two systems. However, when influenced by the same surface type (closed ice) the values are of the same magnitude." We do not expect perfect agreement of the different systems, and it is not our objective to explain differences of each individual flux interval by comparing the flux footprint of the different systems in detail. The surface types assigned in the discussion of our measurements are based on estimates of the uniform fetch upwind of the gradient system. We used the Horst (1999) footprint estimation for gradient and profile flux measurements (based on Horst and Weil, 1992; Horst and Weil 1994), and with the six measurement heights from 0.05 m to 1.29 m, the fetch influencing the profile measurement is typically less than 10 m.

In the revised manuscript, we have now included a brief discussion of how different footprints of the measurements at different height levels of the gradient system can complicate the interpretation of our profile data (cf. our response to reviewer #2):

"It must be noted that the footprint area affecting the lowest measurement heights of the gradient system becomes very small, and even for the narrow lead surface type, the ice surface with low temperatures will influence these measurements."

The uncertainty estimation methodology, though commendable in its use of Monte Carlo simulations, underrepresents potential systematic errors. For example, the correction for tubing losses is based on size distribution measurements from a ship-based DMPS located several meters above the sea surface, which may not accurately represent the near-surface environment sampled by the gradient system. Variability in vertical gradients of particle size, especially under stratified conditions or during local emissions, could lead to an incorrect penetration fraction estimate and a biased flux. A sensitivity analysis showing how variations in assumed size distributions affect the loss correction would greatly improve transparency. Additionally, more details on the impact of inlet length, orientation, and isokinetic sampling conditions should be included to assess sampling biases under varying wind regimes.

We agree that uncertainty estimates of real world measurements can always be extended to include more potential sources of error. For the Monte Carlo simulations, we focus on the influence of uncertainty estimates of the particle number concentration measurement, the temperature measurement and the wind speed measurement. The correction of particle number concentrations for particle losses in the inlet tubing was applied beforehand. We acknowledge that there is a potential for over-correction or under-correction due to several reasons. As stated in the original manuscript, particle size distributions were not measured on the ice, and therefore we have to assume that the particle size distribution data taken from onboard the icebreaker were representative also for the particles sampled on the ice. We also present the variation of the loss correction for all observed particle size distributions in the original manuscript. It is clearly stated that the hourly median penetrating particle fraction varied between 90 % and 98 %. This was meant to give the reader a transparent impression of the sensitivity of the loss correction on the particle size distribution. The inlet length was not changed and is described in the original manuscript. When the gradient mast was set up, the inlet was oriented toward the prevailing wind direction with the opening facing downward. Thus, the inlet was oriented perpendicular to the main flow direction.

The use of Monte Carlo simulations to estimate uncertainty in particle and sensible heat fluxes is a commendable choice; however, the manuscript lacks a sufficiently detailed explanation of how this method was applied specifically in the context of aerosol flux measurements. In aerosol science, uncertainty propagation is particularly sensitive to input data variability, non-linearity in particle losses, and the low signal-to-noise ratios often encountered in Arctic conditions. The authors state that 10,000 random values were generated per profile, but it is unclear whether the distributions used were strictly Gaussian, whether input variances were assumed to be independent across heights, or how temporal autocorrelation in measurement noise was handled. Furthermore, the choice to summarize daily uncertainties using the 90th percentile of simulated fluxes lacks justification—was this percentile empirically chosen, or based on prior studies? Overall, a clearer explanation of the statistical assumptions, potential bias sources, and limits of the method's sensitivity is needed to allow readers to evaluate how well the simulation captures real-world variability and uncertainty in the derived fluxes.

For Monte Carlo simulations, the following steps were taken to prepare the input data: First, the standard deviation of wind speed, temperature and particle number concentration was calculated for each height level based on data sampled every second, and then the median value of these standard deviations was calculated per day. Within this median standard deviation, 10,000 values were chosen randomly from a uniform distribution. We did not use a Gaussian distribution nor the Poisson distribution typically describing counting statistics for single-particle counting detectors such as our CPC but rather the simple and robust uniform distribution with equal probabilities for all values within the range defined by the daily median value auf the standard deviation for the three variables of wind speed, temperature

and particle concentration. Based on these random values, the turbulent flux was recalculated 10,000 times for each profile. To obtain a reasonable uncertainty estimate without the largest outliers, the 90th percentile was chosen following, e.g. Watanabe and Pfeiffer (2022), who used a Monte Carlo approach and presented the 90th percentile as their uncertainty estimate for sea surface temperature. The Monte Carlo simulation section was modified as follows:

Line 216: The standard deviation was calculated for each individual height level; following, the mean standard deviation across all height levels for one day is used.

Watanabe, T. K., & Pfeiffer, M. (2022). A simple Monte Carlo approach to estimate the uncertainties of SST and δ18Osw inferred from coral proxies. Geochemistry, Geophysics, Geosystems, 23, e2021GC009813. https://doi.org/10.1029/2021GC009813

The comparison of friction velocity estimates between the gradient system and eddy covariance (Figure 2) shows substantial scatter, yet the potential causes of these discrepancies are not sufficiently explored. In particular, it is unclear whether inertial motion correction was applied to the eddy covariance measurements on the ship mast. Motion-induced errors are known to bias vertical velocity estimates on moving platforms, and the absence of an inertial measurement unit (IMU) or equivalent correction could lead to systematic over- or underestimation of u*, especially under low turbulence. *I recommend that the authors clarify whether motion correction was applied* and, if not, discuss this as a potential limitation. In addition, the eddy covariance post-processing methods (e.g., coordinate rotation, averaging strategy) should be described more explicitly, and their influence on the u* estimates should be assessed. Without such information, the comparability of the methods remains questionable, particularly over complex surfaces like narrow leads where the surface heterogeneity is likely to exacerbate footprint mismatches.

We understand the concerns expressed in this comment. Motion correction as well as additional corrections to compensate for flow distortion (described in Prytherch et al. 2015) were applied for the eddy covariance data. Eddy covariance calculations for 20 min averaging intervals followed standard procedures including despiking, double coordinate rotation and linear detrending. This information is now included in the revised manuscript:

Line 113: Eddy covariance calculations followed standard procedures including despiking, double coordinate rotation and linear detrending. Motion correction and a correction to compensate for flow distortion were also applied (Prytherch et al., 2015).

Prytherch, J., Yelland, M. J., Brooks, I. M., Tupman, D. J., Pascal, R. W., Moat, B. I., and Norris, S. J.: Motion-correlated flow distortion and wave-induced biases in air–sea flux measurements from ships, Atmos. Chem. Phys., 15, 10619–10629, https://doi.org/10.5194/acp-15-10619-2015, 2015.

In terms of data analysis, the manuscript occasionally uses ambiguous terminology when referring to "net emission" and "net deposition". It is not always clear whether these terms

refer to statistically significant fluxes (i.e., exceeding uncertainty bounds) or to the algebraic sign of the estimated flux. In several figures, flux values that lie within the uncertainty range are still color-coded as deposition or emission. I suggest the authors define a threshold for meaningful flux detection (e.g., based on the daily maximum error estimate) and clearly distinguish between statistically significant fluxes and those near the noise level. This would reduce the risk of overinterpreting marginal cases.

Thank you for this comment. The terms net emission and net deposition always refer to the algebraic sign of the estimated flux. The uncertainties calculated by the Monte Carlo simulation may indicate a significantly higher uncertainty than is present. For this reason, the uncertainty of the particle fluxes is indicated in Figures 5 and 6 and discussed in the text. To give a few examples, the following is stated in line 321 of the original manuscript: "As outlined in Sect. 2.3, the uncertainties of the calculated fluxes can be considerable." And in line 328/329: "However, taking into account the estimated maximum uncertainty (grey shading in Fig. 5), fluxes may be positive or negative but with a tendency toward net particle emission." And in line 373/374: "However, it is possible that net emission fluxes may occur if the maximum estimated uncertainty (grey shading) is taken into account."

Throughout the manuscript, the authors report both absolute particle fluxes in $m^{-2} \cdot s^{-1}$ and normalized fluxes in $cm \cdot s^{-1}$, which is a standard approach in aerosol micrometeorology. However, the manuscript lacks a clear explanation of why and when each form is used. The definition of normalized flux ($V\_D = -P/C$) is provided, but the rationale for expressing it in [cm/s] rather than [m/s] is not discussed. This unit switch can be confusing, particularly since figure axes and captions do not always indicate units explicitly. I recommend that the authors (1) clearly define both flux formats early in the Methods or Data Analysis section, (2) justify the choice of units for normalized flux, and (3) ensure that all figure axes and table entries explicitly label the flux units used. A brief explanation of the comparability advantage of normalized fluxes (especially under variable concentration regimes) would also strengthen the interpretation of results.

As stated in the manuscript, we introduce the normalized flux to allow comparison of particle fluxes independent of varying particle number concentrations and with other studies. Therefore, we present normalized fluxes when discussing the vertical exchange independent of particle number concentration, and when comparing with other studies (e.g. Table 2). For the normalized flux, the unit "cm/s" is used throughout the manuscript without switching units. Similarly, particle fluxes are always presented in units "$m^{-2} s^{-1}$". In the figure titles, the unit has been added everywhere, as for example in Figure 5. We carefully checked all figures to ensure that units are given. As suggested, the units are now introduced in the Methods Section in the revised manuscript.

Line 174: To convert the kinematic sensible heat flux to units W m$^{-2}$ we multiplied the value in units K ms$^{-1}$ with the density of air (1.225 kg m$^{-3}$) and the specific heat capacity (1005 J kg$^{-1}$ K$^{-1}$) of air.

Line 346: Figure 5. Particle flux ($10^6$ m$^{-2}$ s$^{-1}$) at surface type wide lead on (a) 17 May and (b) 18 May 2023 as well as particle flux at surface type narrow lead on (C) 20 May and (d) 21 May 2023. The brown color indicates net particle emission, and the blue color indicates net particle deposition. The grey-shaded bars illustrate the maximum error value for the day of measurement, as estimated by Monte Carlo simulations.

Line 207: To allow comparison of particle fluxes independent of varying particle number concentrations and with other studies, a normalized flux $V_D$ (cm s$^{-1}$), is calculated by Equation 11:

Line 316: To remove the influence of varying particle number concentrations, the normalized flux (in cm s$^{-1}$ e.g. Farmer et al. 2021) calculated according to Equation 11 will also be presented.

While the authors interpret many of the observed concentration and flux patterns in terms of local surface type (wide lead, narrow lead, closed ice), they do not apply any trajectory analysis to objectively evaluate the origin of the sampled air masses. This is a significant limitation. In polar environments, long-range transport can dramatically influence background particle concentrations, size distributions, and chemical composition. The manuscript refers to air mass changes (e.g., "southerly winds bringing humid air") and suggests terrestrial influence from Greenland or Svalbard, but these assertions are speculative without the support of backward trajectory modeling (e.g., HYSPLIT, FLEXPART). Including at least a qualitative or cluster-based trajectory analysis would strengthen the interpretation of aerosol variability and help distinguish between local emission/deposition processes and transported signals. I recommend the authors include such an analysis or, at minimum, acknowledge this limitation in the discussion section.

We agree that air mass origin are valuable when interpreting particle concentrations and size distributions. Air mass back trajectories were calculated for the expedition using the Lagrangian Analysis Tool LAGRANTO (Murto and Tjernström 2024). Any references to air mass changes in the manuscript are based on these backward trajectories. The backward trajectories were also used to subdivide the expedition into six meteorological periods briefly referenced in the original manuscript. A reference to the calculated trajectories has now been added in the revised manuscript.

Line 117: General meteorological conditions during the AoM expedition were categorized in six periods, mainly defined using temperature and humidity profiles from 6-hourly radio soundings as well as 7-day back-trajectories, calculated with the Lagrangian Analysis Tool LAGRANTO on ERA5 data (see also Table 1; Murto et al. 2024, Murto and Tjernström 2024).

Murto, S. and Tjernström, M: Backward trajectories along the ship track using the LAGRANTO model, for the expedition ARTofMELT, Arctic Ocean, 2023. Dataset version 1. Bolin Centre Database. https://doi.org/10.17043/oden-artofmelt-2023-trajectories-backward-ship-1, 2024.

In summary, this manuscript presents an important contribution to the field of Arctic micrometeorology and aerosol flux measurements. The deployment of a mobile gradient flux system under challenging field conditions, combined with extensive observational data across varied surface types, offers valuable insights into particle exchange processes in the central Arctic. However, to ensure scientific rigor and broader relevance, the manuscript requires substantial revision. Key areas for improvement include deeper engagement with prior literature on gradient methods, clarification of methodological details (particularly around uncertainty estimation and regression quality), and more cautious interpretation of flux patterns in the absence of chemical or trajectory data. Provided that these issues are thoroughly addressed, I believe the paper will make a strong and meaningful contribution and would ultimately be worthy of publication.We fully understand that this is a comment in the discussion and that further development of the manuscript may already be underway. Nevertheless, we would be very grateful if the authors would consider addressing at least some of the issues and suggestions raised here. We believe that doing so would significantly enhance the clarity, robustness, and scientific value of this already promising contribution.

Again, we thank P. Markuszewski and M. Mårtensson for their comprehensive and stimulating comments. We hope that we adequately addressed all concerns raised above, and we are convinced that we improved the manuscript based on these comments.

References

Andreas, E. L.: Comment on "Vertical coarse aerosol fluxes in the atmospheric surface layer over the North Polar Waters of the Atlantic" by Tomasz Petelski and Jacek Piskozub, J. Geophys. Res.-Oceans, 112, C11, https://doi.org/10.1029/2007JC004191, 2007.

Held, A., Brooks, I. M., Leck, C., and Tjernström, M.: On the potential contribution of open lead particle emissions to the central Arctic aerosol concentration, Atmos. Chem. Phys., 11, 3093–3105, https://doi.org/10.5194/acp-11-3093-2011, 2011a.

Held, A., Orsini, D. A., Vaattovaara, P., Tjernström, M., and Leck, C.: Near-surface profiles of aerosol number concentration and temperature over the Arctic Ocean, Atmos. Meas. Tech., 4, 1603–1616, https://doi.org/10.5194/amt-4-1603-2011, 2011b.

Markuszewski, P., Nilsson, E. D., Zinke, J., Mårtensson, E. M., Salter, M., Makuch, P., and Piskozub, J.: Multi-year gradient measurements of sea spray fluxes over the Baltic Sea and the

North Atlantic Ocean, *Atmos. Chem. Phys.*, **24**, 11227–11253, https://doi.org/10.5194/acp-24-11227-2024, 2024.

Nilsson, E. D., Rannik, Ü., Swietlicki, E., Leck, C., Aalto, P. P., Zhou, J., and Norman, M.: Turbulent aerosol fluxes over the Arctic Ocean: 2. Wind-driven sources from the sea, J. Geophys. Res.-Atmos., 106, 32139–32154, https://doi.org/10.1029/2000JD900747, 2001.

Petelski, T.: Marine aerosol fluxes over open sea calculated from vertical concentration gradients, J. Aerosol Sci., 34, 359–371, https://doi.org/10.1016/S0021-8502(02)00191-9, 2003.

Petelski, T., and Piskozub, J.: Vertical coarse aerosol fluxes in the atmospheric surface layer over the North Polar Waters of the Atlantic, J. Geophys. Res.-Oceans, 111, C06039, https://doi.org/10.1029/2005JC003295, 2006.

Petelski, T., Piskozub, J., and Paplińska-Swerpel, B.: Sea spray emission from the surface of the open Baltic Sea, J. Geophys. Res.-Oceans, 110, C10023, https://doi.org/10.1029/2004JC002800, 2005.

Savelyev, I. B., Anguelova, M. D., Frick, G. M., Dowgiallo, D. J., Hwang, P. A., Caffrey, P. F., and Bobak, J. P.: On direct passive microwave remote sensing of sea spray aerosol production, Atmos. Chem. Phys., 14, 11611–11631, https://doi.org/10.5194/acp-14-11611-2014, 2014.